# Allosteric effects of the coupling cation in melibiose transporter MelB

Parameswaran Hariharan[1], Yuqi Shi[2], Amirhossein Bakhtiiari[3], Ruibin Liang[3], Rosa Viner[2], Lan Guan[1]*

[1]Department of Cell Physiology and Molecular Biophysics, Center for Membrane Protein Research, School of Medicine, Texas Tech University Health Sciences Center, Lubbock, United States; [2]Thermo Fisher Scientific, San Jose, United States; [3]Department of Chemistry and Biochemistry, Texas Tech University, Lubbock, United States

## eLife Assessment

This manuscript presents **useful** insights into the molecular basis underlying the positive cooperativity between the co-transported substrates (galactoside sugar and sodium ion) in the melibiose transporter MelB. Building on years of previous studies, this **convincing** study improves on the resolution of previously published structures and reports the presence of a water molecule in the sugar binding site that would appear to be key for its recognition, introduces further structures bound to different substrates, and utilizes binding and transport assays, as well as HDX-MS and molecular dynamics simulations to further understand the positive cooperativity between sugar and the co-transported sodium cation. The work will be of interest to biologists and biochemists working on cation-coupled symporters, which mediate the transport of a wide range of solutes across cell membranes.

*For correspondence:
lan.guan@ttuhsc.edu

**Abstract** The major facilitator superfamily (MFS) transporters play significant roles in human health and disease. *Salmonella enterica* serovar Typhimurium melibiose permease ($MelB_{St}$) catalyzes the symport of galactosides with $Na^+$, $H^+$, or $Li^+$ and is a prototype of MFS transporters. We published the structures of $MelB_{St}$ in both inward- and outward-facing conformations, bound to galactoside or $Na^+$, and proposed that positive cooperativity of the co-transported solutes is crucial for the symport mechanism. Here, we elucidated the underlying mechanisms by analyzing $MelB_{St}$ dynamics and the effects of melibiose, $Na^+$, or both using hydrogen-deuterium exchange mass spectrometry (HDX-MS). We also refined the determinants of sugar recognition by solving the crystal structures of a uniporter D59C $MelB_{St}$ complexed with melibiose and other sugars, and by identifying a critical water molecule involved in sugar recognition. Our integrated studies, combining structures, HDX-MS, and molecular dynamics simulations, support the conclusion that sugar-binding affinity is directly correlated with protein dynamics. $Na^+$ acts as an allosteric activator, reducing the flexibility of dynamic residues in the sugar-binding site and in the cytoplasmic gating salt-bridge network, thereby increasing sugar-binding affinity. This study provides a molecular-level framework of the symport mechanism that could serve as a general model for cation-coupled symporters.

## Introduction

The Solute Carrier (SLC) family of transporters encompasses diverse superfamilies of membrane proteins with various protein folds and employing different transport mechanisms to facilitate the translocation of a wide range of solutes across cell membranes (*Ferrada and Superti-Furga, 2022*).

The largest superfamily of SLC transporters is the major facilitator superfamily (MFS; *Pao et al., 1998*), which includes a significant number of cation-coupled secondary active transporters. MFS transporters are responsible for the uptake of a broad spectrum of solutes across cell membranes, playing crucial roles in physiology, pathology, and pharmacokinetics, and are emerging as drug targets (*César-Razquin et al., 2015*; *Lin et al., 2015*). Recent rapid advancements in membrane protein research have greatly enhanced our understanding of protein conformation and mechanisms (*Guan and Kaback, 2006*; *Yan, 2015*; *Drew et al., 2021*; *Guan, 2023*); however, critical details remain lacking. For example, in cation-coupled symport, it is still unclear how the coupling cations facilitate the binding, translocation, and accumulation of the primary substrate.

For MFS secondary active transporters, most members use $H^+$ as the coupling cation, and a few members use $Na^+$, such as the $Na^+$-coupled essential lipid transporter (MFSD2A) that is expressed in the major organ barriers, including the blood-brain barrier or blood-retina barrier (*Nguyen et al., 2014*; *Cater et al., 2021*; *Chua et al., 2023*). The $Na^+$-coupled melibiose transporter of *Salmonella enterica* serovar Typhimurium (MelB$_{St}$), which is a well-characterized representative for the $Na^+$-coupled MFS transporters, catalyzes the symport of a galactopyranoside with $Na^+$, $H^+$, or $Li^+$, and is a valuable model system for studying cation-coupled transport mechanisms (*Wilson and Ding, 2001*; *Maehrel et al., 1998*; *Meyer-Lipp et al., 2006*; *Guan et al., 2011*; *Granell et al., 2010*; *Guan, 2018*; *Ethayathulla et al., 2014*; *Hariharan and Guan, 2017*; *Guan and Hariharan, 2021*; *Hariharan et al., 2024b*; *Hariharan et al., 2024a*). Two major conformations of MelB$_{St}$ have been determined: an outward-facing conformation at the apo or galactoside-bound states (*Guan and Hariharan, 2021*; *Hariharan et al., 2024a*) and an $Na^+$-bound inward-facing conformation (*Hariharan et al., 2024b*). The primary substrate-specificity determinant pocket and the cation-specificity determinant pocket have been structurally and functionally characterized (*Hariharan and Guan, 2017*; *Guan and Hariharan, 2021*; *Hariharan et al., 2024b*; *Hariharan et al., 2024a*; *Katsube et al., 2022*). All three coupling cations compete for the same binding pocket, and the transport stoichiometry is 1 galactoside: 1 cation ($Na^+$, $H^+$, or $Li^+$). Binding of the primary and coupling substrates is positively cooperative; the sugar affinity depends on the cation identity, with the cooperativity numbers (fold of increase in affinity) being 8, 5, or 2 for $Na^+$, $Li^+$, and $H^+$, respectively (*Hariharan and Guan, 2017*; *Guan and Hariharan, 2021*). In addition to being sensitive to the binding of the coupling cation and its identity, notably, the sugar-binding affinity is also dependent on MelB$_{St}$ conformation (*Hariharan et al., 2024b*). By trapping MelB$_{St}$ in an inward-facing state using the inward-facing conformation-specific binder nanobody-725 (Nb725), both experimental sugar-binding assays and cryo-EM structural analysis support that the sugar-binding pocket at the inward-facing conformation is at a low-affinity state (*Hariharan et al., 2024b*; *Katsube et al., 2023*). Remarkably, $Na^+$ binding to the inward-facing conformation remains unchanged (*Hariharan et al., 2024b*; *Katsube et al., 2023*). These results provide experimental evidence supporting the previously proposed stepped-binding kinetic model for melibiose/$Na^+$ symport, in which $Na^+$ binds first and is released after the sugar release on the opposite surface (*Guan et al., 2011*; *Ethayathulla et al., 2014*; *Guan et al., 2012*; *Hariharan and Guan, 2014*).

Positive cooperativity of substrate binding has been proposed to be the key symport mechanism in MelB$_{St}$, but the molecular basis for this critical mechanism remains unclear. The structures indicate that the bound sugar and $Na^+$ have no direct contact, while the two binding pockets are in close proximity (*Guan and Hariharan, 2021*; *Hariharan et al., 2024b*). The minimum free-energy landscape for sugar translocation, which was simulated based on the structures of the outward- and inward-facing conformations as two starting points (*Liang and Guan, 2024*), suggests that the $Na^+$ contribution to the binding free energy of sugar by direct contact is negligible. Cooperativity occurs through allosteric coupling, likely via electrostatic interactions.

In this study, we further analyzed the sugar-binding site by improving the crystal structure resolution of α-NPG bound at 2.60 Å, which uncovered an important water molecule in the binding site. We also confirmed the sugar specificity determinants in both sugar and MelB$_{St}$ by determining the crystal structures in complex with three other α-sugar substrates—melibiose, raffinose, or α-methyl galactoside (α-MG)—that vary in the number of sugar units but all contain an α-galactosyl group. We then examined the structural dynamics of the entire MelB with hydrogen-deuterium exchange coupled to mass spectrometry (HDX-MS) and the effects of melibiose, $Na^+$, or both on MelB$_{St}$. HDX-MS is particularly attractive for dynamic systems, as the provided information yields beyond the stable events within the dynamic repertoire of transporter proteins. By monitoring the HDX rate across a time

window, it can provide predictive modeling for overall protein folds (*Hamuro, 2024*; *Masson et al., 2019*). The resolution of structural information provided by this technique is typically on the peptide level, though. We also performed MD simulations to analyze the side-chain dynamics. Collectively, all results support the notion that the sugar-binding affinity of MelB$_{St}$ is coupled to protein structural dynamics and conformational transitions between inward- and outward-facing states, and the coupling cation in this symporter functions as an allosteric activator.

## Results

### Sugar transport and binding affinity measurements

The sugar analog α-NPG has been determined as a substrate for MelB$_{Ec}$ by measuring *p*-nitrophenol production from the cells that expressed both MelB$_{Ec}$ and α-galactosidase (*Wilson and Wilson, 1987*). The same assay was modified to determine the α-NPG translocation mediated by MelB$_{St}$ in DW2 cells. Melibiose at 1 mM was added to induce α-galactosidase expression during cell growth, and the washed cells were applied to measure the time course of *p*-nitrophenol release into the media upon adding 1 mM α-NPG into the induced cells. The detection was for the *p*-nitrophenol, which resulted from α-NPG transport and hydrolysis. The results showed that both WT MelB$_{St}$ and D59C uniport mutant mediated the translocation of α-NPG (*Figure 1a*; *Figure 1—source data 1a*) at this downhill mode of transport.

To determine raffinose transport activity, [³H]raffinose transport in the *E. coli* DW2 strain was carried out in the absence of Na⁺ and Li⁺ or the presence of Na⁺ or Li⁺ (*Figure 1b*; *Figure 1—source data 1b*). The results showed that raffinose, a trisaccharide formed from melibiose and fructose, is also a substrate and transported by MelB$_{St}$.

Previously, our ITC studies in the presence of Na⁺ revealed that the binding affinity, $K_d$ values, of the WT and D59C MelB$_{St}$ for melibiose were 1.25 mM ± 0.05 mM and 4.96 ± 0.11 mM, respectively, and for α-NPG were 16.46 ± 0.21 μM or 11.97 ± 0.09 μM, respectively (*Hariharan and Guan, 2017*; *Guan and Hariharan, 2021*; *Hariharan and Guan, 2014*; *Hariharan and Guan, 2021*). The same assay was utilized to determine the α-MG and raffinose binding in the presence of Na⁺ (*Figure 1c–d*; *Figure 1—source data 1c-d*). By injecting the α-MG or raffinose in a buffer-matched solution into the sample cell containing the WT MelB$_{St}$ or the D59C uniporter mutant in the presence of Na⁺, the isotherm curve fitted well, yielding $K_d$ values of 0.88 ± 0.04 mM or 3.79 ± 0.53 mM, for α-MG or raffinose, respectively. Raffinose, which has more sugar units, exhibits a poor binding affinity to MelB$_{St}$.

### Outward-facing crystal structures of MelB$_{St}$ complexed with varied sugar substrates

The uniporter D59C MelB$_{St}$ mutant exhibits greater thermostability, making it a valuable tool for structural analysis of sugar binding (*Guan and Hariharan, 2021*). Here we report four crystal structures of D59C MelB$_{St}$ with the endogenous sugar substrate melibiose, and two other α-galactosides, methyl α-galactoside (α-MG) with a single sugar unit or raffinose with three sugar units, as well as α-NPG at an improved resolution. The structure statistics were presented in *Supplementary file 1*. All structures adopt a virtually identical outward-facing conformation with RMSD values of less than 0.4 (*Figure 2—figure supplement 1*). This typical MFS-fold transporter with 12 transmembrane helices is organized into two six-helix domains linked by the middle loop between helices VI and VII (Loop$_{6-7}$). There are three cytoplasmic helices at the loop$_{6-7}$, loop$_{8-9}$, and the C-terminal tail or lid (*Figure 2*), named ICH1-3, respectively; and ICH1 and ICH2 run parallel to the membrane bilayer. In all structures, one sugar molecule is bound in the middle of the protein and sandwiched by both N- and C-terminal domains as described previously (*Guan and Hariharan, 2021*). As indicated by the sliced surface, the binding residues located within the cytoplasmic leaflet of both domains define the edge of the inner barrier, which prevents the sugar from passing across the transporter into the cytoplasm. On the periplasmic side, the open vestibule connects the solvent to the binding pocket. In the α-NPG-bound structure, as reported in PDB ID 8FRH (D59C apo structure) or ID 8FQ9 D55C mutant with DDMB (*Hariharan et al., 2024a*), a PEG molecule was modeled to the density near Arg296 along with helix IX, which could indicate a potential lipid-binding site (*Figure 2—figure supplement 1*).

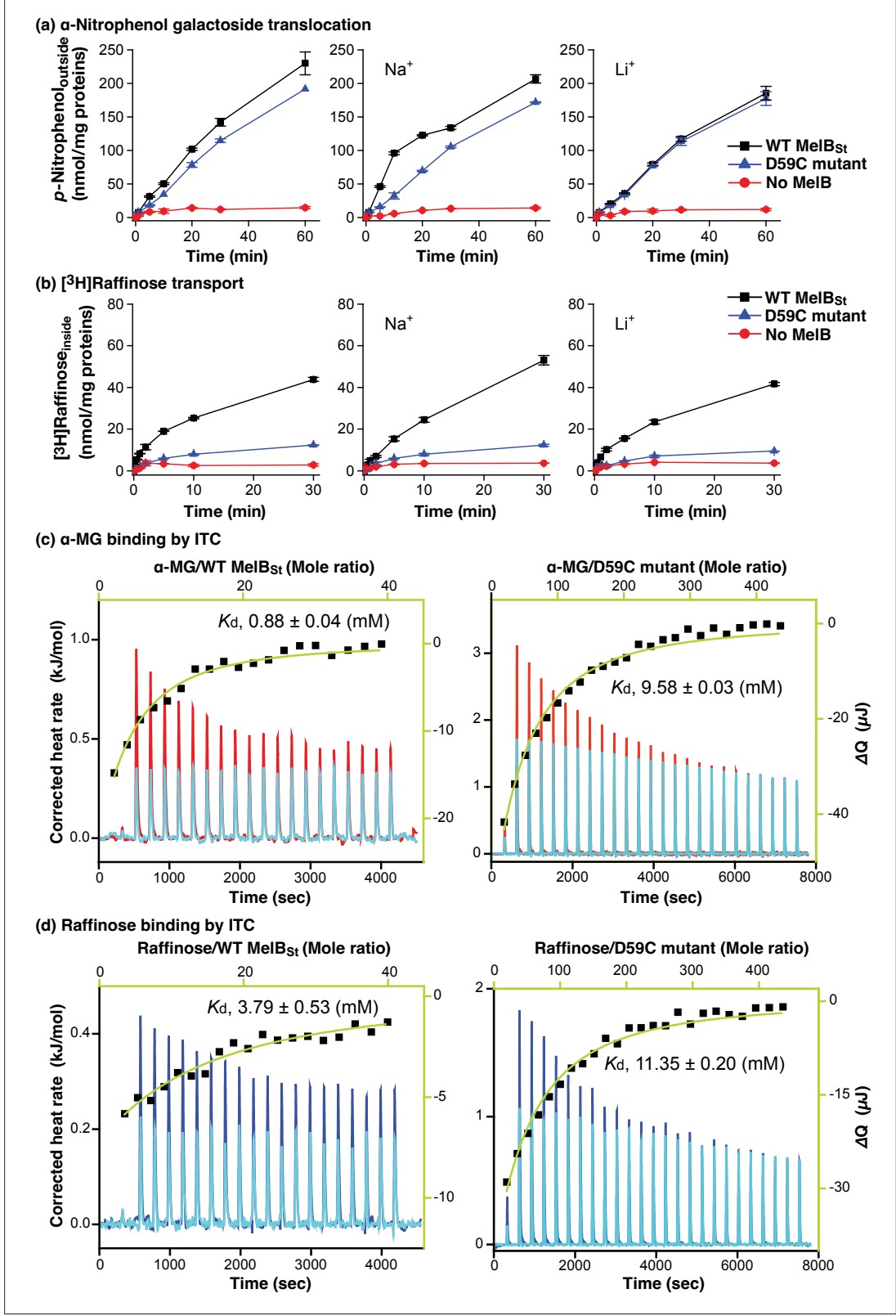

**Figure 1.** Functional characterizations. (**a**) α-NPG downhill transport. *E. coli* DW2 cells with the induced α-galactosidase by melibiose in the absence or presence of WT MelB$_{St}$ or D59C mutant were washed before incubating with 0.5 mM *p*-nitrophenyl-α-D-galactoside at 30 °C in the absence or presence of 20 mM NaCl or 20 mM LiCl as described in Methods. The cell-aliquots at 0, 1, 5, 10, 20, 30, and 60 min were quenched with 0.3 M Na$_2$CO$_3$, followed by centrifugation to remove the cells. The *p*-nitrophenol in the supernatant released from the cells was measured at $A_{405}$ nm. The mean values from two

*Figure 1 continued on next page*

*Figure 1 continued*

tests were plotted against incubation time with standard error bars. (**b**) [³H]Raffinose active transport. The *E. coli* DW2 cells in the absence or presence of MelB$_{St}$ with no α-galactosidase induction were used for the active transport of [³H]raffinose at 1 mM (specific activity, 10 mCi/mmol) at 23 °C in the absence or presence of 50 mM NaCl or LiCl. The cellular uptake time course measurements at 0, 0.08, 0.17, 0.5, 1, 2, 5, 10, and 30 min were carried out by a dilution and fast-filtration method. The mean values from two tests were plotted against the incubation time with standard errors. (**c & d**) ITC measurement of α-MG (**c**) or raffinose (**d**). ITC measurements were performed at 25 °C under similar buffer conditions: 20 mM Tris-HCl, pH 7.5, 100 mM NaCl, 10% glycerol, and 0.035% UDM detergent. For each experiment, 80 µM of the purified WT MelB$_{St}$ or D59C mutant was placed in the reaction cell, and methyl α-D-galactoside (α-MG) or raffinose at 10 mM (against the WT, left column) or 100 mM (against D59C, right column) from the syringe was incrementally titrated to generate the thermograms. The curve fitting is performed with a one-site independent-binding model included in the NanoAnalyze software (version 3.7.5). The thermograms were plotted as baseline-corrected heat rate (µJ/sec; left axis) vs. time (bottom axis) for the titrant to MelB$_{St}$ (red for α-MG and blue for raffinose) or to buffer (light blue). The heat change ΔQ (µJ; filled black symbol) was plotted against the mole ratio of the sugar to MelB$_{St}$ (top/right axes in green). The $K_d$ values were the average of two tests with standard error. (**c**) α-MG. (**d**) Raffinose. Source data are available for panels a-d as *Figure 1—source data 1a–d*.

The online version of this article includes the following source data for figure 1:

**Source data 1.** Excel tables for transport and ITC curves.

## α-NPG-binding structure refined to a resolution of 2.60 Å

The improved resolution, from 3.01 Å to 2.60 Å, provided a better-resolved density map for the bound α-NPG molecule, which further supported the originally assigned pose, as demonstrated by the stereo view (*Figure 3*; *Guan and Hariharan, 2021*). Thus, the binding pocket is formed by 14 residues on five helices, including N-terminal residues, as labeled in black including helices I (Lys18, Asp19, Ile22, and Tyr26), IV (Tyr120, Asp124, and Tyr128), and V (Arg149 and Ala152), and the C-terminal residues labeled in blue including helices X (Trp342) and XI (Gln372, Thr373, Val376, and Lys377).

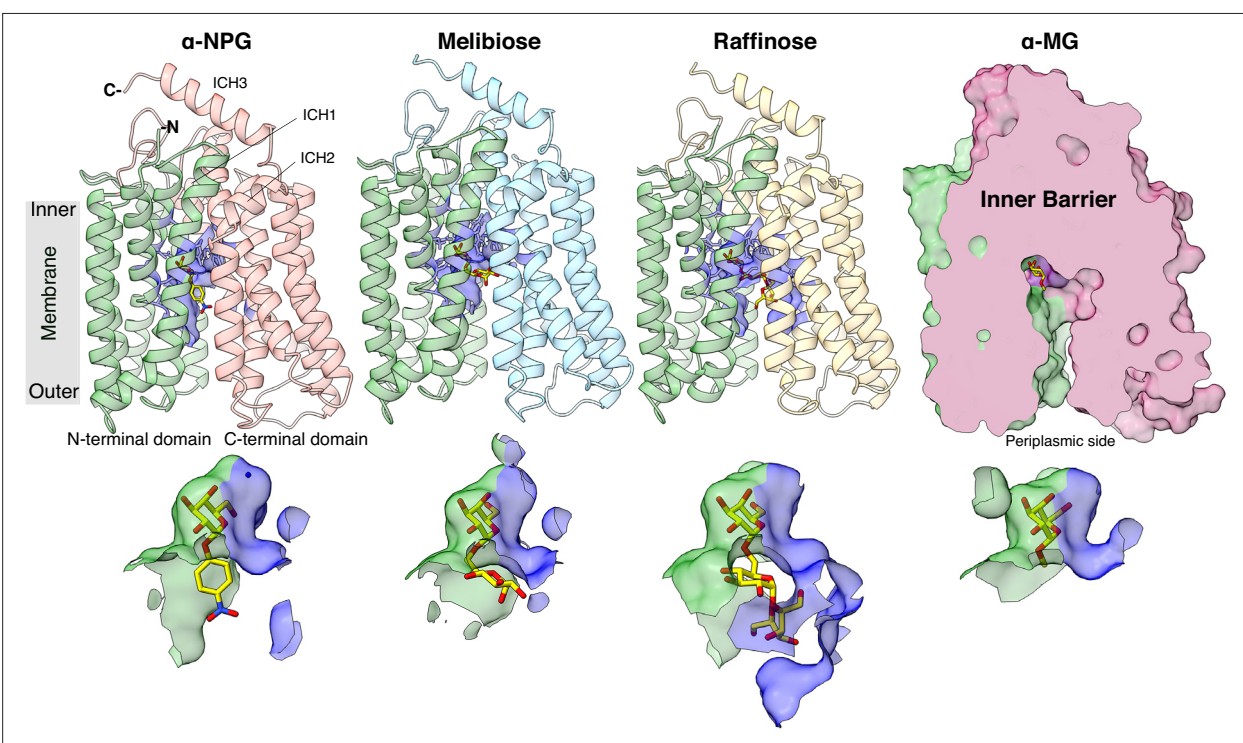

**Figure 2.** Crystal structures of D59C MelB$_{St}$ in complex with α-sugars substrates. *Upper row*: Cartoon representation of the structures of D59C MelB$_{St}$ bound with α-NPG, melibiose (α-disaccharide), and raffinose (α-trisaccharide), respectively, along with a surface presentation of D59C MelB$_{St}$ complexed with α-methyl galactoside (α-MG). All structures were oriented with the cytoplasmic side on top and the N-terminal domain (colored green) on the left. Each sugar molecule is colored yellow. The blue sticks and surfaces indicate residues within 5 Å of the sugar molecules. *Lower row*: Sugar-binding pocket. Residues from the N-terminal and C-terminal domains were shown in surface representation and colored in green and blue, respectively.

The online version of this article includes the following figure supplement(s) for figure 2:

**Figure supplement 1.** Overlay.

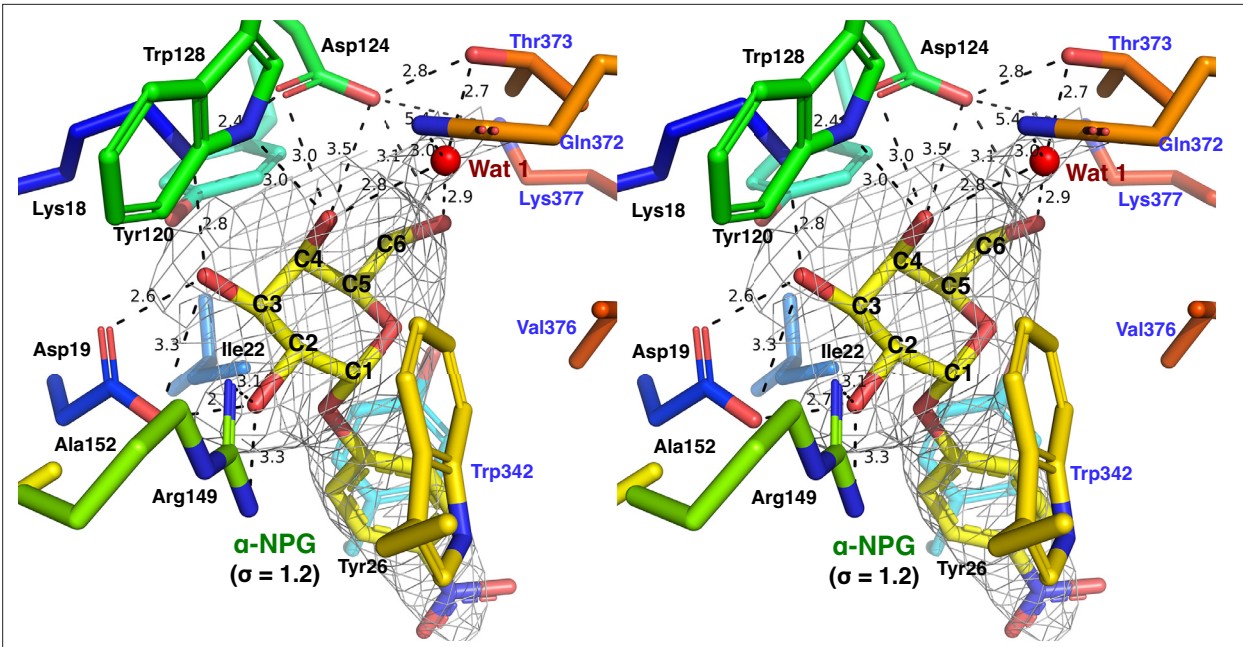

**Figure 3.** α-NPG binding. The cross-eye stereo view of α-NPG binding. The side chains forming the binding pocket formed from N- and C-terminal domains were shown in stick representation, colored according to corresponding hosting helices in rainbow, and labeled in black and blue, respectively. Isomesh map of the α-NPG (in yellow) and Wat 1 (in red) was contoured at a level of σ=1.2. Dashed lines indicate distances within hydrogen-bonding or salt-bridge interactions (Å).

The online version of this article includes the following figure supplement(s) for figure 3:

**Figure supplement 1.** Waters interacting with the salt-bridge network.

A water molecule (Wat 1) was modeled to a positive density, which is located at hydrogen-bonding distances from the C4-OH and C6-OH on the galactopyranosyl ring and the Thr373 at helix XI, and also surrounded by Gln372 at helix XI and Asp124 at helix IV, as shown in the stereo-view of a 2Fo-Fc electron density map (*Figure 3*). Another three water molecules participated in the cytoplasmic gating salt-bridge network between both bundles (*Figure 3—figure supplement 1*). Water molecules Wat 2 and Wat 3 interacted with the charged pair Arg295 (helix IX) and Asp351 (helix XI), respectively, and Wat 4 interacted with Lys138 from loop$_{4-5}$, which also formed a salt-bridge interaction with Glu142 in this functionally important gating area (*Figure 3—figure supplement 1b*; *Amin et al., 2014*).

## Melibiose binding

The endogenous substrate melibiose is formed from a galactose unit and a glucose unit linked by an α–1,6 galactosyl bond (D-Gal-(α1→6)-D-Glc). The melibiose-bound D59C MelB$_{St}$ structure was refined to a resolution of 3.05 Å (*Figure 4a and e*), and the density map displayed a clear two-unit blob, fitting well with one molecule of melibiose. Two water molecules, Wat 2 and Wat 3, were modeled; however, no strong peak was observed at the Wat 1 position. Notably, the binding affinity between melibiose and α-NPG differs by a factor of 100; in α-NPG, the hydrophobic phenyl group, positioned at 4 Å distance from the phenyl group of Tyr26, forms a strong stacking interaction. While melibiose has four additional OH groups, the glucosyl ring sits nearly perpendicular to the phenyl ring of Tyr26 (*Figure 4—figure supplement 1*) with no strong polar interaction with the transporter.

## Methyl α-D-galactoside (α-MG) binding

α-MG has a methyl substituent at the anomeric C1 position of the galactosyl moiety in an α-linkage. The crystal structure of D59C MelB$_{St}$ complexed with α-MG was refined to a resolution of 3.68 Å, and the density map clearly displayed a one-unit blob in the binding pocket, where one α-MG was modeled similarly to the galactosyl moiety of melibiose (*Figure 4b*).

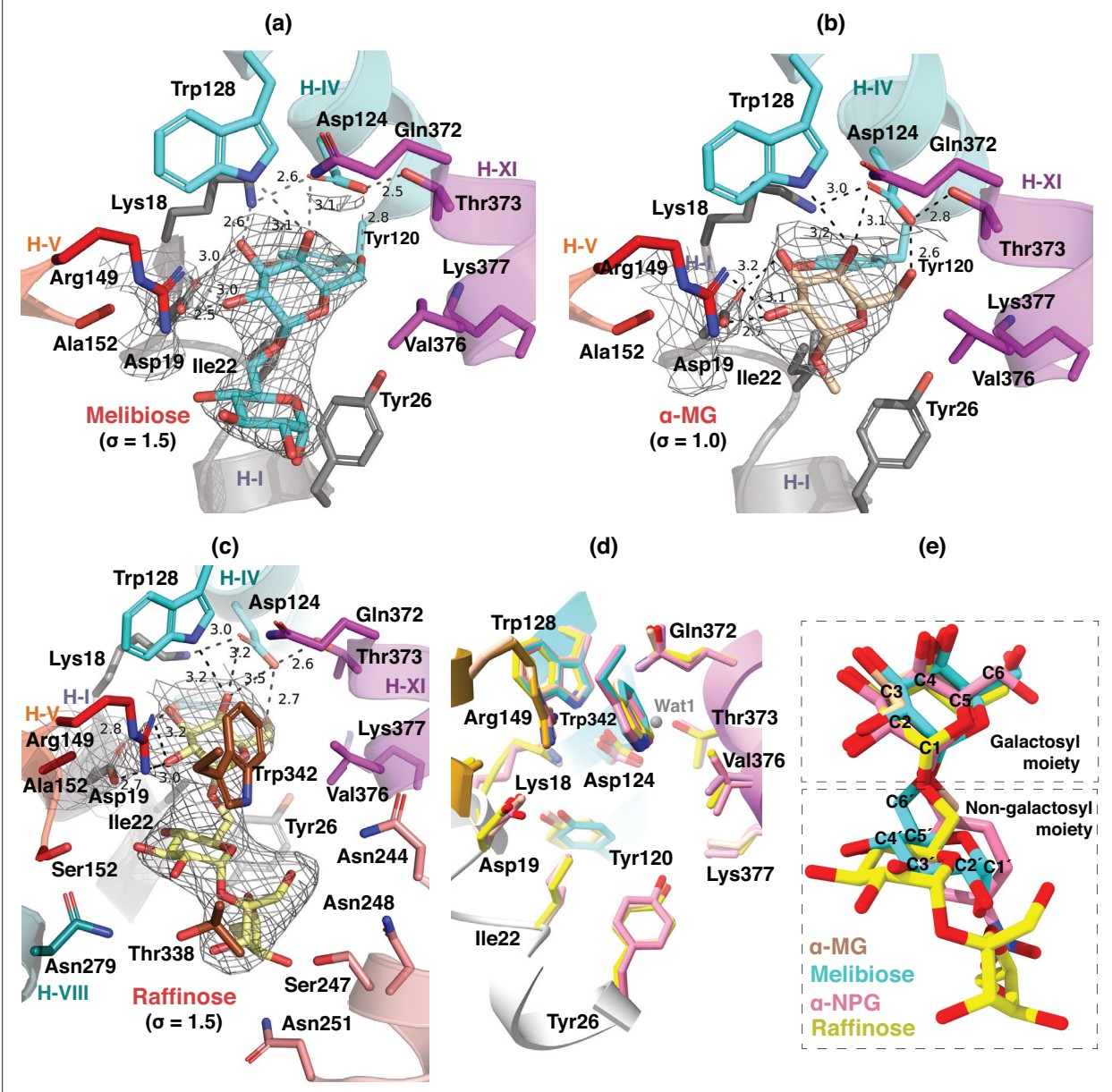

**Figure 4.** Binding of melibiose, α-MG, and raffinose. All residues within a 5 Å distance to the bound substrates were shown in sticks. Dashed lines, the distances within hydrogen-bonding and salt-bridge interactions (Å). (**a**) Melibiose binding. (**b**) α-MG binding. (**c**) Raffinose binding. (**d**) Residues in the sugar-binding pockets from the alignment of all four structures. (**e**) Substrates from the alignment of all four structures. Carbon positions on the galactosyl (C1-6) and glucosyl moiety (C1′–6′) were labeled on the melibiose molecule. Trp342 was removed for clarity in panels a, b, and d. Isomesh maps for each sugar and Asp19 were contoured at levels of σ=1.5 for melibiose and raffinose or σ=1.0 for α-MG.

The online version of this article includes the following figure supplement(s) for figure 4:

**Figure supplement 1.** Overlay of the bound melibiose and α-NPG.

## Raffinose binding

The raffinose-bound D59C MelB$_{St}$ mutant structure was refined to a resolution of 3.40 Å, and the density map displayed a clear three-unit blob in the binding pocket, fitting well with this trisaccharide. As expected, the raffinose-binding pocket is significantly larger than that of melibiose and α-NPG (*Figure 4c*), involving two additional helices. Consequently, a set of polar side chains on helix VII (Asn244, Ser247, Asn248, and Asn251), helix VIII (Asn279), and helix V (Ser153) were within 4–5 Å distances to either the glucosyl or fructosyl moieties. Interestingly, the polar interactions from raffinose to MelB$_{St}$ are limited to the galactosyl moiety, which is nearly identical to those presented

in α-MG, melibiose, or α-NPG, while the Asp19 is at a hydrogen-bonding distance from the OH-4 on the glucosyl moiety.

All four galactosyl moieties align well in the specificity-determinant pocket, regardless of the number of monosaccharide units (*Figure 4d and e*; *Figure 2—figure supplement 1*), which provided a strong structural support to the previous conclusion that the galactosyl moiety determines the specificity of the primary substrates and the non-galactosyl moiety contributes to the binding affinity.

## Structural dynamics measured by HDX-MS

To determine the structural dynamics of MelB$_{St}$ and the influence of melibiose and/or Na$^+$, the differential HDX-MS experiments were employed between MelB$_{St}$ alone and MelB$_{St}$ with melibiose and/or Na$^+$ (*Masson et al., 2019*; *Jia et al., 2020*; *Zmyslowski et al., 2022*; *Supplementary file 2*). MelB$_{St}$ residues 2–470 out of 476 were covered, achieving a coverage of over 87.47% and 86.62% from 149 or 152 overlapping deuterium-labeled peptides (*Supplementary files 2 and 3*; *Figure 5—figure supplement 1*, *Figure 5—source data 1 and 2*). There are 59, 63, or 59 non-covered residues for the datasets of the apo vs. melibiose-bound, Na$^+$-bound, and melibiose- and Na$^+$-bound states, respectively. Most are at the transmembrane helices IV, IX, X, and XI, and some cytoplasmic loops. Labeled peptides cover all periplasmic loops.

### Six regions with greater deuteration at the apo state

The peptide coverage-based deuteration per residue plot and the relative deuterium uptake per peptide plot were generated from mean values from all three time points of each peptide with six duplicates (*Figure 5a and b*; *Supplementary file 3*). The data indicate that, aside from the N- and C-terminal tails, which exhibit high HDX, six regions showed greater deuterium uptake in the apo state. As highlighted in red labels, two were in the N-terminal helices (I and V) and four were in the C-terminal loops (loop$_{6-7}$, loop$_{8-9}$, loop$_{9-10}$, and loop$_{10-11}$), which are the dynamic regions of MelB$_{St}$ in the apo state. Notably, all the ICH1-3 were in the dynamic area. Comparing with helices I and V carrying the sugar-binding residues, helices II and IV—housing both the Na$^+$-binding residues (Asp55, Asn58, and Asp59) and part of the sugar-binding residues (Asp124 and Trp128)—exhibited significantly lower deuteration levels.

### The effects of substrate(s) on MelB$_{St}$ dynamics

The differential deuterium labeling (ΔD), which was calculated based on deuterium uptake in the absence (apo state) or presence of specific ligand(s) (Holo state), was shown as the residual plot of each peptide at three labeling time points and the sum of all (*Figure 6a*). The dashed lines indicate the global thresholds calculated for each dataset. A hybrid significance analysis was used to determine the significance: ΔD>the global threshold of each dataset and p<0.05 (*Hageman and Weis, 2019*). More than 50% of residues (237, 264, or 257 positions) exhibited either insignificant ΔD values (D$_{Mel - Apo}$ < |0.186|, ΔD$_{(Na+) - Apo}$ < |0.224|, or ΔD$_{Na(+)Mel - Apo}$ < |0.175|), or p>0.05, respectively (*Supplementary file 2*). Melibiose binding induced a wide range of effects on HDX, including a few protections (less deuterium uptake in the Holo state) and more deprotections (greater deuterium uptake in the Holo state). In contrast, Na$^+$ alone or with melibiose primarily caused protections. The results supported the previous conclusion determined by the thermal denaturation study detected by circular dichroism spectroscopy (*Hariharan and Guan, 2021*); that is the melibiose-, or Na$^+$-bound MelB$_{St}$ was more stable than the apo state, and when MelB$_{St}$ bound with both, it was the best. All deprotected peptides with significance were labeled individually (*Figure 6a*). A peptide with at least one significant ΔD (meeting both criteria) from any time points was mapped onto the melibiose-bound structure (*Figure 6b–d*; *Figure 5—figure supplement 1b*).

Consensus effects were observed in all three labeling conditions (*Figure 6a–d*), and peptides with significant effects were clustered in several regions, which largely overlapped with the dynamic regions obtained at the apo state (*Figure 5*). Four out of six dynamic regions showed protection. Nearly full-length helices I and V, and regions 213–226 covering ICH1, showed protection by melibiose and/or Na$^+$. The peptides covering 364–374 at the loop$_{10-11}$ and the starting part of helix XI, which contains the sugar-binding residues Gln372 and Thr373, were protected by Na$^+$ or Na$^+$ with melibiose. Another two dynamic regions exhibited diverse effects. The cytoplasmic positions 292–298 covering the loop$_{8-9}$ and the starting helix IX, containing a conformation-important residue Arg295, were deprotected by

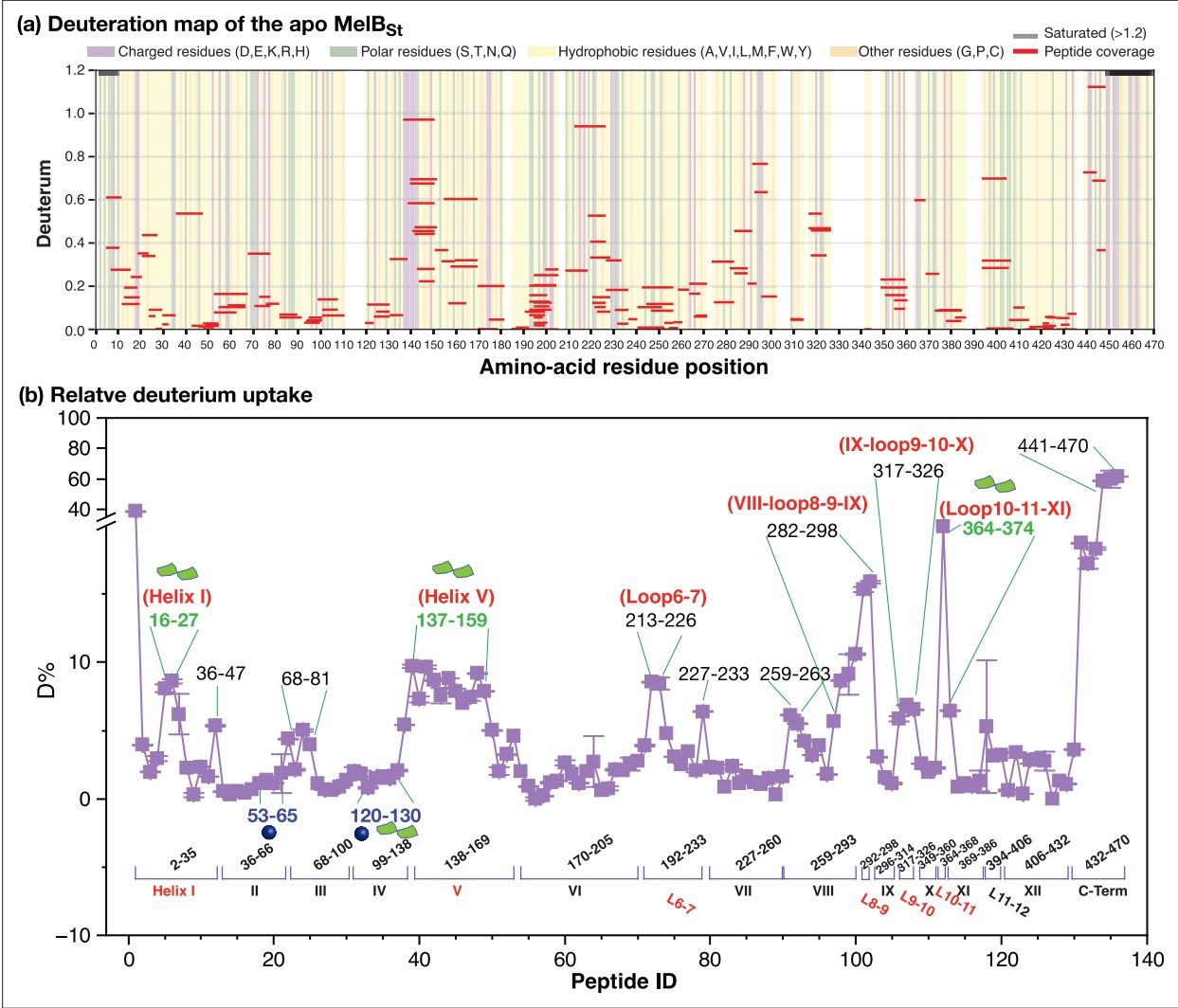

**Figure 5.** Deuterium uptake of the apo MelB$_{St}$. HDX experiments on WT MelB$_{St}$ were conducted as described in Methods. All values were from the mean of six measurements at the apo state. (**a**) Deuteration map of the apo MelB$_{St}$. Mean deuteration levels of MelB$_{St}$ peptides at the apo state averaged first across timepoints (30 s, 300 s, 3000 s), then across two replicates, were presented against amino-acid residue sequence. The values at both N- and C-terminal regions greater than 1.2 are shown as black bars. The chemical properties of the peptide-covered residue were indicated by background shading; the white background indicated the non-covered positions. (**b**) Relative deuterium uptake per peptide plot. The corresponding transmembrane helices and a few loops were marked. Notably, due to the nature of overlapping peptides, the indicated amino acid position is not sequential. Peptides with greater deuterium uptake were labeled. Peptides covering sugar- and cation-binding sites were colored in green and blue, respectively, and the dynamic regions with greater uptakes were colored in red. Source data are available as **Figure 5—source data 1 and 2**.

The online version of this article includes the following source data and figure supplement(s) for figure 5:

**Source data 1.** HDX_MS raw data_apo & Na.

**Source data 2.** HDX-MS raw data - apo, mel, mel&Na.

**Figure supplement 1.** Mapping of HDX data on an inward-facing melibiose-bound D59C MelB$_{St}$.

melibiose but protected by melibiose and Na$^+$. The neighboring peptide 284–291, corresponding to helix VIII and loop$_{8-9}$ covering ICH2 at the cytoplasmic gating area, was also deprotected at the 30 sec time point but protected at the 3000 sec time point by melibiose and Na$^+$ (**Figure 6d**). The positions 317–326 at helix IX-loop$_{9-10}$-helix X showed deprotections by melibiose but were protected by Na$^+$. Five out of six loops, except for loop$_{5-6}$ (**Figure 6a–b**, *pink and red*), were deprotected by melibiose. In addition, the lid ICH3 was protected by melibiose or melibiose with Na$^+$, with a single result of deprotection by Na$^+$. The following sections will focus on the sugar- and cation-binding pockets as well as the structural elements critical for conformational transitions. All peptides with ΔD values greater

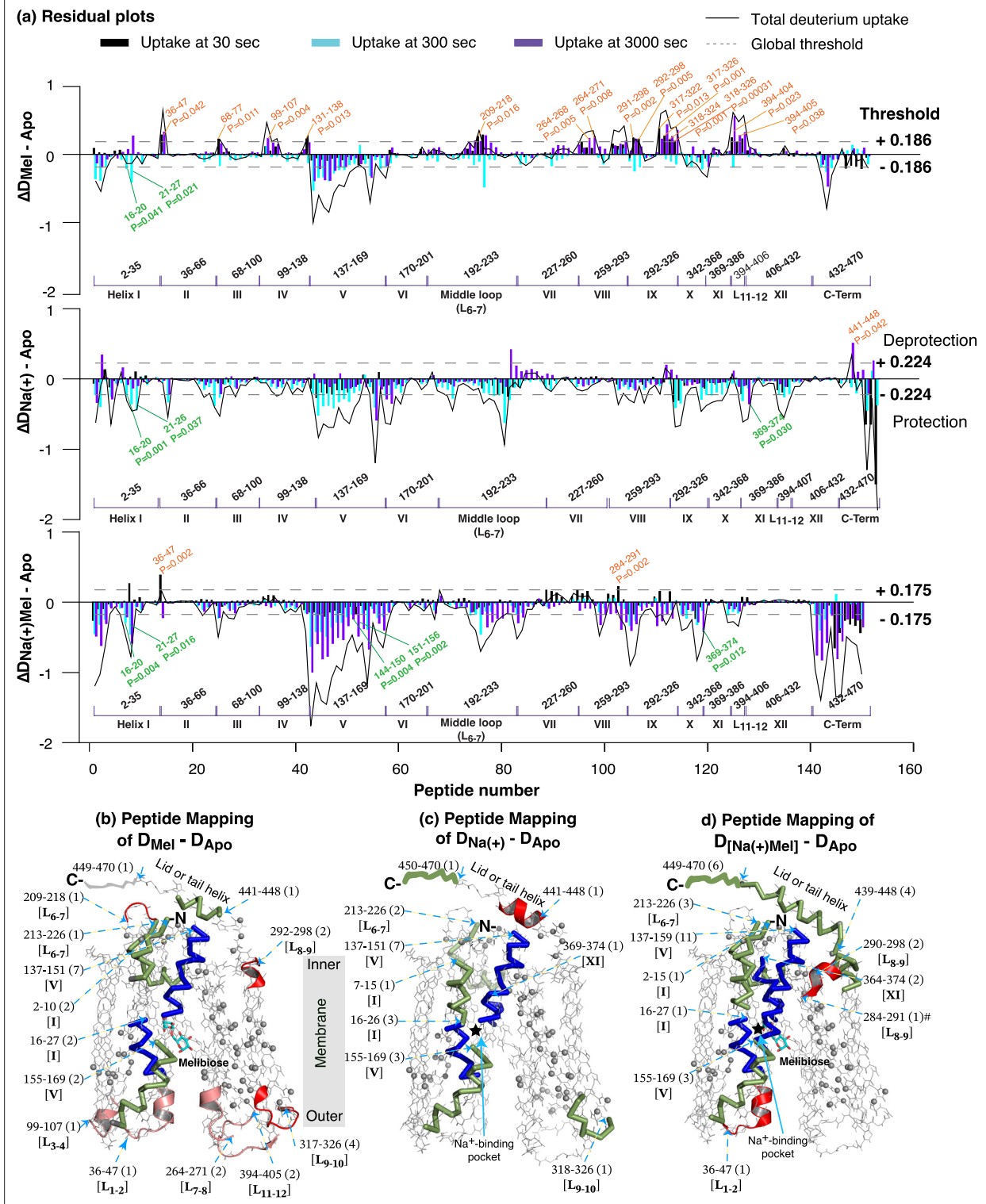

**Figure 6.** Residual plots and structural mapping. HDX experiments on WT MelB$_{St}$ in the apo or holo states (with melibiose, Na$^+$, or Na$^+$ plus melibiose) were conducted as described in Methods. (**a**) Residual plots (D$_{Holo - Apo}$). Differential deuterium uptakes (ΔD) of each time point and the total uptake calculated from paired conditions for position 2–470 were plotted against the peptide number. Black, cyan, and purple bars, the deuterium uptake at 30, 300, and 3000 s, respectively; dark gray curve, total uptake from all three time points. Deprotection, ΔD$_{Holo - Apo}$ > 0; protection, ΔD$_{Holo - Apo}$ < 0. Each sample was analyzed in triplicate. Dashed lines indicate the levels of the global threshold values calculated from each dataset, as labeled. The protein residue positions corresponding to the overlapping peptides were marked, and the covered transmembrane helices were labeled in Roman numerals.

*Figure 6 continued on next page*

*Figure 6 continued*

All deprotected peptides with statistical insignificance (ΔD>threshold and p<0.05) in each dataset were labeled. (**b–d**) Peptide mapping on the crystal structure of the melibiose-bound inward-facing conformation for D$_{Mel – Apo}$, D$_{Na(+) – Apo}$, and D$_{Na(+)Mel – Apo}$, respectively. The red star symbol indicated the location of the Na$^+$-binding pocket, the drawing lines represented the disordered MelB$_{St}$ C-terminal tail, and a gray bar showed the membrane region of MelB$_{St}$. The peptides of ΔD values with statistical significance (ΔD > |threshold| and p<0.05) at any timepoint are highlighted either in ribbon representation for protection (colored in blue for peptides covering sugar-binding pocket and in green for all other regions) or in cartoon representation for deprotection (colored pink for data from 3000 s and red for data from 30 s). Non-covered residues from each dataset are shown as gray spheres at the Cα position and listed in *Supplementary file 3*, and peptides with statistically insignificant differences (either ΔD < |threshold| or p>0.05) are illustrated in backbone representation in gray. Peptide positions are marked by their starting residue; the number of overlapping peptides is indicated in round brackets, and the location is shown in square brackets.

The online version of this article includes the following figure supplement(s) for figure 6:

**Figure supplement 1.** Deuterium uptake time course of all peptides ΔD value >threshold.

**Figure supplement 2.** Deuterium uptake time course of all peptides ΔD value >threshold.

**Figure supplement 3.** Deuterium uptake time course of all peptides ΔD value >threshold.

than the threshold value were selected, and their deuterium uptake time course plots were presented (*Figure 6—figure supplements 1–3*).

## A flexible sugar-binding pocket with a rigid cation-binding pocket

This HDX study covered most positions for the binding pockets for sugar and Na$^+$ (*Supplementary file 4*). Ten representative peptides were highlighted by uptake time course plots with structural mapping (*Figure 7*). Three peptides 53–62 (helix II) and 120–123 and 121–130 (helix IV), which cover all Na$^+$-binding residues (Asp55, Asn58, Asp59, and Thr121), exhibited lower deuteration levels at the apo state (*Figures 5 and 7*, *black open squares*) with no significant effect by melibiose and/or Na$^+$

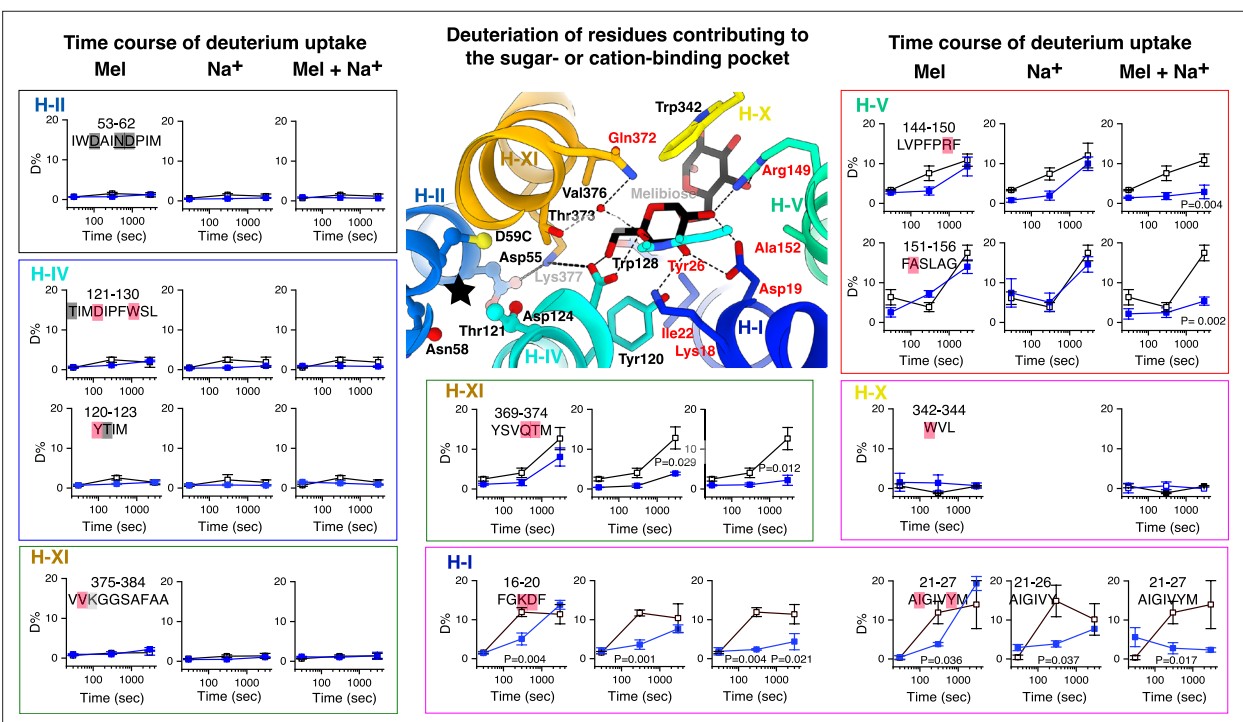

**Figure 7.** Allosteric effects on substrate-binding sites. Deuterium uptake time course of representative peptides covering the sugar- and cation-binding pockets. The percentage of deuterium uptake measured in the absence (empty black square) or presence of melibiose (Mel), Na$^+$, or Mel and Na$^+$ (filled blue square) was plotted against labeling times of 0, 30, 300, and 3000 s. The peptide sequences were shown with residues at the cation- or sugar-binding pocket highlighted in black or red, respectively. Lys377 between the two binding pockets was highlighted in gray. p values are provided for each time point where the ΔD value exceeds the threshold. On the melibiose-bound structure, residues participating in the sugar binding and cation binding were shown in stick. black star, the cation-binding pocket; red text labels: residues showing greater HDX with significant substrate effects; black text labels: residues showing poor HDX with no significant substrate effects.

(*Figure 7*, *blue filled squares*). Others in helices II and IV also exhibited similar behavior (*Figure 5*; *Supplementary file 3*), indicating that both helices, including the Na⁺-binding residues, are conformationally rigid.

As described, helices I and V are the major flexible transmembrane helices (*Figure 5*), which cover the six sugar-binding residues (Lys18, Asp20, Ile22, Tyr26, Arg149, and Ala152). Peptides 16–20 and 21–27 (helix I) were significantly protected by melibiose, Na⁺ that binds remotely, or both (*Figure 7*). Arg149 (helix V) was well covered by several overlapping peptides that consistently demonstrated greater protections by melibiose and/or Na⁺ (*Figures 6 and 7*). The shorter peptide 144–150 (helix V) showed protection by melibiose or Na⁺ alone, which is statistically significant; however, the magnitude of the change is subtle. In the presence of both, the protection became significant. The Ala152-carrying peptide showed significant protection by melibiose and Na⁺, and peptide 368–373, which covers the sugar-binding residues Gln372 and Thr373 (helix XI), also exhibited significant protection by Na⁺ alone or in combination with melibiose. Peptides carrying other sugar-binding residues Asp124 and Tyr128 (helix IV), and Trp342 (helix X) and Val376/Lys377 (XI) showed poor deuteration and no significant effects by either melibiose, Na⁺, or both. Overall, the results indicated that the sugar-binding residues in proximity to the cation-binding pocket are rigid with no significant effect by either substrate; in contrast, peptides carrying the sugar-binding residues far from the cation-binding site are dynamic, and their flexibility was significantly inhibited by sugar binding itself, by Na⁺ alone, especially by the binding of both, which supports that the melibiose affinity is correlated with the conformational flexibility and Na⁺ can increase the sugar binding by inhibiting the conformational dynamics.

## Dynamics of the cytoplasmic gating salt-bridge network and modulations by substrates

The cytoplasmic gating salt-bridge network between the two domains, involving nine charged residues (*Figure 8a and c*), was most covered. Residues Lys138, Arg141, and Glu142 at loop$_{4-5}$-helix V are three critical positions from the N-terminal domain. Overlapping peptides that covered this region consistently showed greater deuteration levels at the apo state and greater protections under all three conditions, as represented by the peptide 137–150 (*Figures 5–7*). Notably, the sugar-binding residue Arg149 at the gate area of helix V is within this most dynamic area of the transmembrane domain of MelB$_{St}$. Another gating residue, Arg295 (helix IX), which forms multiple interactions in this network, including two water molecules (*Figure 3—figure supplement 1*), was covered by peptide 292–298, which showed a higher deuteration level and was deprotected by melibiose but protected by melibiose and Na⁺. Peptide 364–368 at Loop$_{10-11}$ and the connected region of helix XI, which contains Glu365 at the other side of this salt-bridge network, also showed high deuterium uptake and protection by melibiose with Na⁺. On the contrary, Asp341, Asp354, and Glu357 at the rigid helix X, which are located in the center of this salt-bridge network, showed poor deuteration under all conditions. The results showed that gating residues surrounding the rigid helix X in this broadly spanned cytoplasmic salt-bridge network are highly dynamic and sensitive to the binding of melibiose, Na⁺, and melibiose with Na⁺.

As shown by the aligned outward- and inward-facing conformations and the membrane topology (*Figure 8b–c*), two gating regions at ICH2 and helix I-loop$_{1-2}$-helix II, covered by C-terminal peptide 284–291 and N-terminal peptide 36–47 at the cytoplasmic or periplasmic side, respectively, were deprotected by melibiose or together with Na⁺. In addition, another gating region at loop$_{11-12}$ covered by the C-terminal peptide 394–405 and the peptide 318–326 at helix XI-loop$_{9-10}$-helix X was also deprotected by melibiose. Most deprotected areas are at the periplasmic loops.

## Molecular dynamics (MD) simulations

Three systems, including the apo, melibiose-bound, and melibiose- and Na⁺-bound states of a generated WT MelB$_{St}$, were constructed by embedding in a POPE:POPG (7:2) lipid bilayer. For each system setup, five independent replicas of MD simulations were carried out, each for ~400 ns, and the total sampling time for each system was ~2 µs.

### Water 1 occupancy

To analyze the Wat 1 modeled in the higher resolution structure with α-NPG bound, a distance-based criterion for defining the average water occupancy in the sugar-binding site across all trajectories of

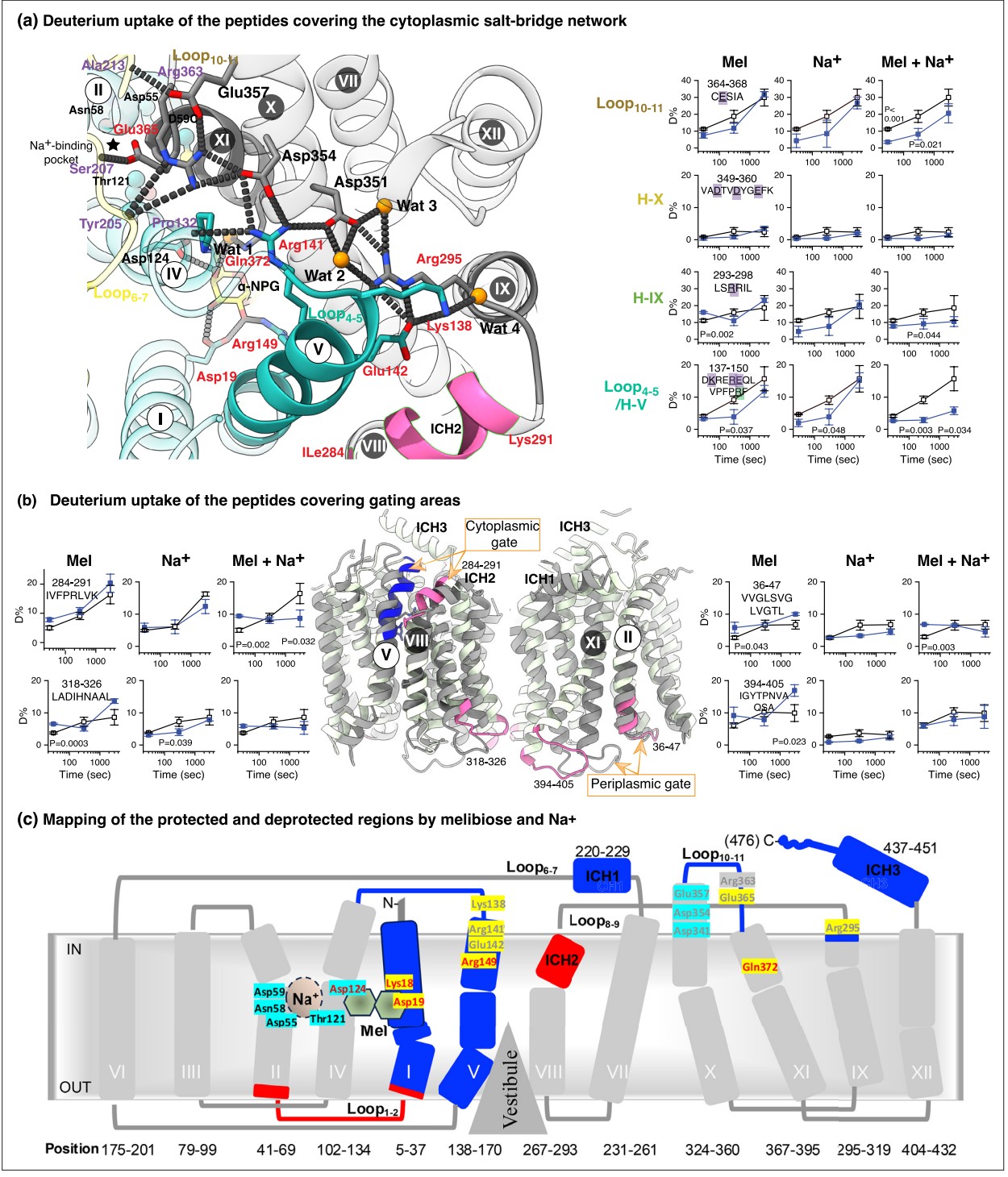

**Figure 8.** Allosteric effects on gating salt-bridge network. (**a**) Deuterium uptake time course of representative peptides covering the cytoplasmic gating salt-bridge network. The percentage of deuterium uptake measured in the absence (empty black square) or presence of melibiose (Mel), Na⁺, or Mel and Na⁺ (filled blue square) was plotted against labeling times of 0, 30, 300, and 3000 s. p values are provided for each time point where the ΔD value exceeds the threshold. The peptide sequences were shown with residues at the salt-bridge network or sugar-binding pocket highlighted in purple or red, respectively. On the α-NPG bound structure, the gating salt-bridge network Lys138, Arg141, and Glu142 (Loop₄₋₅/IV), Arg295 (IX), Asp351, Asp354, and Glu357 (**X**), and Arg363 and Asp365 (Loop₁₀₋₁₂) and their polar contacts with the backbone of other positions were shown in dashed lines. The N- and C-terminal domains were colored in light cyan and gray, respectively. The residues at the cation binding pocket were shown in ball and stick, as also indicated by the red star. Wat, water. Three sugar-binding residues, Asp19, Arg149, and Gln272, were shown in stick. Arg363 is in the list of uncovered positions. The region covering both the cytoplasmic gating salt-bridge network and sugar-binding residue Arg149 was highlighted in solid color, and

*Figure 8 continued on next page*

*Figure 8 continued*

the peptide 284–291 at helix III-ICH2 was also highlighted in pink. (**b**) Deprotection at loops. Deuterium uptake time course of four peptides at loops, mainly at the gating area, was presented and also mapped on the outward-facing structure, which was overlayed with the inward-facing structure [PDB 8T60]. The cytoplasmic and periplasmic gates were indicated. The peptides with deprotection by substrate were colored in pink. Model at the left side, helices V and VIII in front; model at the right side, helices II and XI in front. (**c**) HDX and ligand effects mapping on an outward-facing topology model of MelB$_{St}$. Melibiose- and Na$^+$-binding residues were labeled in red or black, respectively, and residues in the gating salt-bridge network were labeled in gray. Residues with higher levels of deuteration and ligand-induced protections were highlighted in yellow background, and residues with low levels of deuteration and no ligand effects were colored in cyan background. This topological figure was modified from the Figure 3—figure supplement 4 of the article 'Mobile barrier mechanisms for Na$^+$-coupled symport in an MFS sugar transporter' published by eLife (*Hariharan et al., 2024b*).

The online version of this article includes the following figure supplement(s) for figure 8:

**Figure supplement 1.** The gating and sugar-binding residue Arg149.

each system as described in Methods. Results showed that Wat-1 exhibited nearly full occupancy when melibiose was present, regardless of whether Na$^+$ was bound at the cation-binding site (*Supplementary file 6*), which supported the crystal structure observation.

## Side-chain flexibility

The side-chain heavy-atom root-mean-square fluctuation (RMSF) over five replicas of trajectories was analyzed for the apo and MelB$_{St}$ with melibiose and Na$^+$ bound states (*Figure 9*). The results showed that the side chains in the cation-binding site and nearby sugar-binding positions reduced fluctuations upon the binding of substrates. The residues Asp19, Tyr26, Arg149, and Gln372 at helices I, V, and XI were relatively flexible, and their conformational freedoms were significantly reduced by the binding of melibiose and Na$^+$. The data about the side chain flexibility are consistent with the peptide-based HDX results.

## Discussion

### The binding recognition and affinity of MelB$_{St}$ have been further refined

Dehydration of sugar molecules is expected for binding; however, full or partial dehydration to bind MelB is unknown. The 2.60 Å resolution α-NPG-bound structure showed a partially dehydrated sugar at the binding site. The bound water connected the OH-4 and OH-6 on the galactopyranosyl ring with Thr373 and Gln372 at helix XI, and it was also surrounded by Asp124 at helix IV. Previous Cys-scanning mutagenesis has shown that the T373C mutant retained most activities. Still, a Cys residue on the Gln372 position significantly decreased the transport initial rate, accumulation, and melibiose

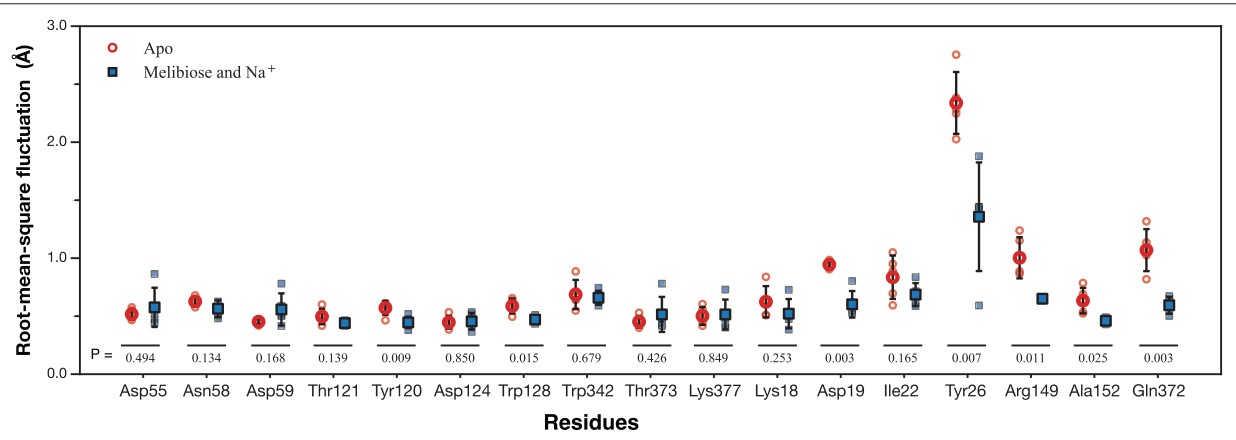

**Figure 9.** Side-chain flexibility analyzed from MD trajectories. For each residue, the side-chain RMSF values in each of the five replicas for the apo and melibiose- and Na$^+$-bound states were plotted. The mean of the RMSF values in each state is represented as a red circle (apo state) or blue square (melibiose- and Na$^+$-bound state). For each residue, unpaired t-test of the RMSF mean values between the apo and the melibiose- and Na$^+$-bound states was performed and the p-values were presented.

fermentation, with little effect on protein expression (*Markham et al., 2021*), which supported the role of Wat-1 in binding. Notably, the orientation of OH-4 is crucial for distinguishing between galactose and glucose. Previously, the OH-4 has been shown at hydrogen-bonding distances from the carboxyl group of Asp124 and the indole group of Trp128 at helix IV (*Guan and Hariharan, 2021*; *Hariharan et al., 2024a*), and the identification of Wat-1 in this study added the polar residues Gln372 and Thr373 on helix XI to stabilize the critical OH-4 (*Figure 3*). Thus, these interactions define the specificity of galactosyl molecules, and Wat-1 is likely part of sugar binding. Notably, the OH-4 and OH-6 and Wat 1 are in close proximity to the Na$^+$-binding residue Thr121 and the cation-binding important residue Lys377, implying that the Wat-1 is between the two specificity-determining pockets. As shown previously, the OH-3 and OH-2 at the opposite edge of the galactopyranosyl ring form multiple hydrogen-bonding interactions with the charged residues Asp19 and Arg149 at helices I and V, which can be assigned to play a crucial role in stabilizing the recognition of OH-4 and enhancing the binding affinity.

## Dynamics of MelB$_{St}$ at different regions

HDX-MS is a powerful technique that simultaneously discloses the dynamic information in various areas of a protein. The qualitative method overcomes the drawbacks of most site-specific labeling-based techniques; however, it only provides the conformational dynamic information on the equilibrium of all ensembles. In the current study, we determined the deuterium uptake rates of the full-length MelB$_{St}$ in the absence or presence of melibiose and/or Na$^+$, identified major dynamic regions (*Figure 5*), as well as substrate-induced effects on the substrate-binding sites and structural elements critical for conformational transition, including the cytoplasmic salt-bridge network and the gating areas (*Figures 6–8*).

Our studies of HDX complemented with molecular dynamics simulations on side-chain fluctuations showed that the Na$^+$-binding pocket and the sugar-binding residues near the Na$^+$-binding site at helix IV are conformationally rigid since they exhibited low deuteration level for both unbound and bound states (*Figures 5–7 and 9*; *Supplementary files 3 and 4*), which are consistent with the previous HDX-MS results of conformation transition (*Hariharan et al., 2024b*). Those rigidities are likely derived from their hosting helices (II and IV). In addition, the Na$^+$-binding affinity, different from sugar binding, exhibited little difference between the inward- and outward-facing states (*Hariharan et al., 2024b*). The conformational freedom of the cation-binding site and most sugar recognition positions were likely restricted in most states to facilitate sugar binding.

The sugar-binding residues, located away from the Na$^+$-binding site, are dynamic, exhibiting higher deuterium levels and side-chain fluctuations. Most of those positions are within the six dynamic regions identified from the apo state, including Lys18, Asp19, Ile22, Tyr26, Arg149, Ala152, Gln272, and Thr373 at helices I, V, and XI. Those regions were protected to varying extents by melibiose or Na$^+$, and the protection was significantly greater when bound to both substrates (*Figures 5–7*). Even their conformational flexibility was restrained by melibiose and Na$^+$, but the deuterium uptakes at those regions were still greater than those of peptides covering the cation-binding site (*Figure 7*). Notably, substrate-induced protection was also detected in other areas of the hosting helices I and V, as further discussed below.

## Arg149, a sugar-binding residue in the major dynamic gating area

Abundant peptides covering Arg149 consistently showed protection by melibiose or Na$^+$, especially when both were present (*Figures 5–8*). Structurally, Arg149 is near to a stretch of charged residues as shown by this peptide 137-<u>DK138RER141E142</u>QLVPFP**R**F-150 (*Figures 5–7*), where the Lys138, Arg141, and Glu142 have been determined to play critical roles in the functionally crucial cytoplasmic salt-bridge network (*Figure 8*). The dynamics of Arg149 are likely modulated by this charged network and belong to the same dynamics group, which provided the structural basis for coupling between the sugar-binding affinity and protein dynamics. At the inward-open structure, this network is deformed. Arg149, as part of this gate, exhibits a large displacement due to an otherwise steric collision with helix X at the outward-facing state (*Figure 8—figure supplement 1*). The sugar-binding affinity at this inward-facing state was reduced by ~30-fold due to the broken binding pocket and the displacement of Arg149 and Gln372 (*Hariharan et al., 2024b*). Functionally, the single-site R149C mutant inhibited but did not eliminate the melibiose transport (*Ethayathulla et al., 2014*), and single-Cys149 (*Markham et al., 2021*) also fermented melibiose with intact cells. The neighboring P148L mutant

decreased transport $V_{max}$ with a better $K_m$ value (*Jakkula and Guan, 2012*). All data showed that the dynamics of this charged network and sugar-binding affinity are interconnected.

This cytoplasmic gating salt-bridge network has a rigid center dictated by helix XI and dynamic surroundings from helices V, IX, XI, and loops 4–5, 8–9, 6–7, 10–11, and those dynamic positions were significantly inhibited by melibiose, $Na^+$, especially the binding of both. Notably, Nb725 binding to the inward-facing conformation also inhibited the dynamics of the similar regions, as shown by peptides 137–150, 284–289, and 364–368 (*Hariharan et al., 2024b*). The similar HDX profiles from all testing conditions suggested the protein motions followed a similar pathway of the conformational transition. Thus, the dynamics of Arg149 are linked to motion and conformational transitions, as well as sugar-binding affinity. Connectively, it is postulated that $Na^+$ binding between helices II and IV stabilizes the dynamics of the gating salt-bridge network and the flexibility of Arg149 and Asp19, which are located more than 10 Å away, thereby allosterically switching the primary substrate-binding site to the higher-affinity state.

The binding of sugar further stabilizes this network and promotes $Na^+$ binding by preventing $Na^+$ from leaving the cation-binding pocket, thus increasing $Na^+$-binding affinity (*Ethayathulla et al., 2014*; *Liang and Guan, 2024*). Therefore, the cooperative binding of sugar and $Na^+$ is part of the substrate-induced conformational allostery, and all functions together.

There are a few regions with substrate-induced deprotections. Two gating areas were deprotected by melibiose and $Na^+$, including the cytoplasmic ICH2 and $Loop_{1-2}$, which suggested the structural arrangements. Notably, ICH2 is linked with Arg295 at one end of the salt-bridge network, and the dynamics of this area could influence the stability of the gating network, which might trigger the separation and conformational transition to the inward-open state. Thus, the increased dynamics of $loop_{1-2}$ and $Loop_{8-9}$ at both gating areas by melibiose, especially with $Na^+$, can be interpreted as the tendency to form a transition-competent conformation with closing at the periplasmic gate and opening at the cytoplasmic gate. A region at the $loop_{6-7}$ responding to the ligand binding identified by the atomic force microscopy was not observed in this HDX study (*Blaimschein et al., 2023*).

The symporter $H^+$-coupled lactose permease LacY and xylose permease XylE also show protonation dependence of sugar-binding affinity (*Jia et al., 2020*; *Smirnova et al., 2008*; *Grytsyk et al., 2017*), with separate cation- and sugar-binding sites (*Guan and Kaback, 2006*; *Guan et al., 2007*; *Mirza et al., 2006*; *Kumar et al., 2014*; *Sun et al., 2012*). Both transporters favored inward-facing conformation, and substrate binding induced outward-facing conformation, as indicated by the extensive HDX analysis with XylE (*Jia et al., 2020*; *Jia et al., 2023*) and a bunch of biophysical measurements of LacY (*Smirnova et al., 2011*). Notably, several dynamic regions in $MelB_{St}$ were also identified in XylE; both proteins showed binding-induced structural changes in their $loop_{1-2}$ and $loop_{8-9}$ (*Jia et al., 2023*). In $MelB_{St}$, while no quantified information on the ensembles, all experimental results and molecular simulations on the minimum free-energy landscape for sugar translocation suggested that the apo MelB favors the outward-facing conformation, and the outward-facing conformation is further preferred when bound with melibiose and $Na^+$ (*Guan and Hariharan, 2021*; *Hariharan et al., 2024b*; *Liang and Guan, 2024*).

## Summary

Our studies on structure, HDX, and MD simulations allow us to conclude that the sugar-binding affinity of $MelB_{St}$ is coordinated with protein structural motions and conformational transition between inward- and outward-facing states. $Na^+$ binding restrains the dynamics of remote sugar-binding residues via stabilizing the dynamic cytoplasmic salt-bridge network, thereby increasing sugar-binding affinity allosterically. The conclusion provides insightful knowledge to understand cooperative binding and symport mechanisms.

## Materials and methods

### Key resources table

| Reagent type (species) or resource | Designation | Source or reference | Identifiers | Additional information |
|---|---|---|---|---|
| Strain, strain background (*Escherichia coli*) | DW2 | *Botfield and Wilson, 1988* | $melA^+$ $melB^-$ $lacZ^-Y$ | |

*Continued on next page*

| Reagent type (species) or resource | Designation | Source or reference | Identifiers | Additional information |
|---|---|---|---|---|
| Recombinant DNA reagent | pK95/ΔAH/WT MelB$_{St}$/CHis$_{10}$ | *Guan et al., 2011* | | Protein expression |
| Recombinant DNA reagent | pK95/ΔAH/D59CMelB$_{St}$/CHis$_{10}$ | *Ethayathulla et al., 2014* | | Protein expression |
| Chemical compound, drug | Melibiose | Acros Organics (Thermo Fisher Scientific) | Cat# 125375000 | Crystallization |
| Chemical compound, drug | [$^3$H]Melibiose | Perkin-Elmer | Radiolabeled | |
| Chemical compound, drug | [$^3$H]Raffinose | American Radiolabeled Chemicals (ARC) | Radiolabeled, Cat# ART 0229 | |
| Chemical compound, drug | Raffinose | Research Products International Corp., (RPI) | Cat# R20500 | Crystallization |
| Chemical compound, drug | α-Methyl galactoside (α-MG) | Sigma-Aldrich | Cat# M1379 | Crystallization |
| Chemical compound, drug | *p*-Nitrophenyl α-D-galactoside (α-NPG) | Sigma-Aldrich | Cat# N0877 | Crystallization |
| Chemical compound, drug | Undecyl-β-D-maltopyranoside (UDM) | Anatrace | Cat# U300 | |
| Chemical compound, drug | Dodecyl-β-D-maltopyranoside (DDM) | Anatrace | Cat# D310 | |
| Chemical compound, drug | *E. coli* lipids | *Avanti Polar Lipids, Inc* | Extract Polar, Cat# 100600 | Crystallization |
| Chemical compound, drug | Polyethylene glycol 400 (PEG400) | Hampton Research | Cat# HR2-603 | Crystallization |
| Commercial assay or kit | Micro BCA Protein Assay | Pierce Biotechnology, Inc | | |
| Software, algorithm | ccp4i2 program | *Potterton et al., 2018* | X-ray data reduction and analysis | |
| Software, algorithm | Phenix (1.21) | *Liebschner et al., 2019* | Molecular replacement and structure refinement | |
| Software, algorithm | Coot (0.9.8.96)_ | *César-Razquin et al., 2015* | Model building | |
| Software, algorithm | BioPharma Finder software (v 5.1) | Thermo Fisher Scientific | MS data process | |
| Software, algorithm | HDExaminer | Sierra Analytics | HDX-MS data process | |
| Software, algorithm | MATLAB (In-house script) | Thermo Fisher Scientific | HDX-MS data process | |
| Software, algorithm | Pymol (3.1.5.1) | *Schrodinger, 2013* | Molecular visualization program | |
| Software, algorithm | UCSF ChimeraX (1.10) | *Pettersen et al., 2021* | Molecular visualization program | |
| Software, algorithm | Origin 2024 | | Graphical software | |
| Software, algorithm | AMBER24 software package | *Case et al., 2023* | MD simulation | |

## Reagents

Melibiose was purchased from *Acros Organics* (*Thermo Fisher Scientific*). [$^3$H]Raffinose and label-free raffinose were purchased from American Radiolabeled Chemicals (ARC), Inc and Research Products International (RPI) Corp., respectively. Raffinose (RPI Chemicals), α-methyl galactoside (α-MG), and *p*-Nitrophenyl α-D-galactoside (α-NPG) were purchased from Sigma-Aldrich. Detergents undecyl-β-D-maltopyranoside (UDM) and dodecyl-β-D-maltopyranoside (DDM) were purchased from *Anatrace*. *E. coli* lipids (Extract Polar, 100600) were purchased from *Avanti Polar Lipids, Inc* All other materials were reagent grade and obtained from commercial sources.

## Strains and plasmids

*E. coli* DW2 cells (*melA$^+$B$^-$*, *lacZ$^-$Y$^-$*) (*Botfield and Wilson, 1988*) were used for protein expression and functional studies. The expression plasmids pK95/ΔAH/WT MelB$_{St}$/CHis$_{10}$ (*Guan et al., 2011*) and pK95/ΔAH/D59CMelB$_{St}$/CHis$_{10}$ (*Ethayathulla et al., 2014*) were used for constitutive expression.

## MelB$_{St}$ protein expression and purification

Cell growth for the large-scale production of WT MelB$_{St}$ or D59C MelB$_{St}$ was carried out in *E. coli* DW2 cells (*Ethayathulla et al., 2014*; *Pourcher et al., 1995*). Briefly, MelB$_{St}$ purification by cobalt-affinity chromatography (Talon Superflow Metal Affinity Resin, Takara) after extraction by 1.5% UDM. MelB$_{St}$ protein was eluted with 250 mM imidazole in a buffer containing 50 mM NaPi, pH 7.5, 200 mM NaCl, 0.035% UDM, and 10% glycerol, and further dialyzed to change the buffer conditions accordingly.

## Protein concentration assay

The Micro BCA Protein Assay (Pierce Biotechnology, Inc) was used to determine the protein concentration.

## α-NPG transport

MelB$_{St}$-mediated α-NPG downhill transport was detected by the release of the intracellular α-NPG hydrolytic product, *p*-nitrophenol, from *E. coli* DW2 cells expressing high-turnover number α-galactosidase and recombinantly expressed MelB$_{St}$ . *E. coli* DW2 cells carrying the constitutive expression plasmid for the WT MelB$_{St}$ or D59C MelB$_{St}$ uniporter mutant in LB media containing 100 mg/L ampicillin were grown in a 37 °C shaker as described (*Guan et al., 2011*). The overnight cultures were diluted fivefold into fresh LB broth and 100 mg/L ampicillin and then shaken at 30 °C for 4 hr. The expression of α-galactosidase was induced by adding 5 mM melibiose for 1 hr before harvesting the cells for the transport assay. The melibiose-induced cells were washed with 100 mM KP$_i$, pH 7.5, three times to remove the remaining melibiose and Na$^+$, and adjusted to $A_{420}$ of 10 (~0.7 mg proteins/ml) in the assay solution of 100 mM KP$_i$, pH 7.5, 10 mM MgSO$_4$, and 1 mM DTT in the absence or presence of 20 mM NaCl or LiCl. The cells under each condition were equilibrated at a 30 °C incubator for 10 min before the α-NPG transport assay, which was initiated by mixing 0.5 mM label-free α-NPG and incubating for 60 min. A 100 µL cell aliquot at the given time point of 0, 1, 5, 10, 20, 30, 60 min was quenched with 900 µL 0.33 M Na$_2$CO$_3$, followed by a centrifugation at 10,000 × *g* for 5 min to collect the clarified supernatant for absorbance measurements. The extracellular *p*-nitrophenol released from the intracellular hydrolytic product of the translocated α-NPG, which was supplied in the extracellular environment, was measured at 405 nm by a UV spectrometer. The concentration of *p*-nitrophenol is estimated based on its molar extinction coefficient of 18,000 M$^{-1}$cm$^{-1}$.

## [³H]Raffinose transport

The raffinose active transport was carried out by [³H]raffinose uptake with DW2 cells expressing MelB$_{St}$ without the induction of α-galactosidase as described for the melibiose transport assay (*Guan et al., 2011*). Briefly, the cells expressing the WT MelB$_{St}$ or D59C MelB$_{St}$ uniporter mutant, or without a plasmid, were mixed with 2 µL of 25 mM [³H]raffinose (specific activity 10 mCi/mmol) to 50 µL of cells final raffinose concentration at 1 mM in the absence or presence of 50 mM NaCl or LiCl, and the transport reaction was quenched at the given time points and followed by a fast filtration.

## Isothermal titration calorimetry

All ITC ligand-binding assays were performed with the TA Instruments (Nano-ITC device) as described (*Hariharan and Guan, 2017*), which yields the exothermic binding as a positive peak. The MelB$_{St}$ was dialyzed overnight with assay buffer containing 20 mM Tris-HCl (pH 7.5), 100 mM NaCl, 10% glycerol, and 0.035% UDM. The ligands are prepared by dissolving in the same batch of dialysis buffer for buffer matching. In a typical experiment, the titrand (MelB$_{St}$) placed in the ITC Sample Cell was titrated with the specified titrant raffinose or α-methyl galactoside (placed in the Syringe) in the assay buffer by an incremental injection of 2 or 2.5 µL aliquots at an interval of 250 or 300 s at a constant stirring rate of 250 rpm (nano-ITC). MelB$_{St}$ protein samples were buffer-matched to the assay buffer by dialysis. The normalized heat changes were subtracted from the heat of dilution elicited by the last few injections, where no further binding occurred, and the corrected heat changes were plotted against the mole ratio of the titrant to the titrand. The values for the binding association constant ($K_a$) were obtained by fitting the data using the one-site independent-binding model included in the NanoAnalyze software (version 3.7.5). The dissociation constant ($K_d$)=1/$K_a$.

## Crystallization, native diffraction data collection, and processing

The D59C MelB$_{St}$ was dialyzed overnight against the sugar-free dialysis buffer (20 mM Tris-HCl, pH 7.5, 100 mM NaCl, 0.035% UDM, and 10% glycerol), concentrated with Vivaspin column at 50 kDa cutoff, and stored at –80 °C. A phospholipid stock solution of 20 mM was prepared by dissolving the *E. coli* Extract Polar (Avanti, 100600) with a dialysis buffer containing 0.01% DDM. The protein sample was diluted to a final concentration of 10 mg/ml with the same sugar-free dialysis buffer, supplemented with phospholipids at 3.6 mM and 30 mM of melibiose or α-MG, 40 mM raffinose, or 6 mM of α-NPG in DMSO solution. Crystallization trials were conducted using the hanging-drop vapor-diffusion method at 23 °C by mixing 2 μL of protein with 2 μL of reservoir solution. Crystals from D59C MelBSt protein with the melibiose, α-MG, or α-NPG appeared against a reservoir consisting of 100 mM Tris-HCl, pH 8.5, 100 mM NaCl$_2$, 50 mM CaCl$_2$, and 32–35% PEG 400. For the raffinose-containing sample, the crystals were collected from 100 mM Tris-HCl, pH 8.5, 50 mM CaCl$_2$, 50 mM BaCl$_2$, and 32.5% PEG 400. All crystals were frozen in liquid nitrogen within 2 weeks and tested for X-ray diffraction at the Lawrence Berkeley National Laboratory ALS beamlines 5.0.1 (for the melibiose, α-MG, and α-NPG-containing complex datasets) or 5.0.2 (for the raffinose-containing datasets) using the remote data collection method.

ALS auto-processing XDS or DIALS programs output files were further reduced by AIMLESS in the ccp4i2 program for the structure solution (*Potterton et al., 2018*). The statistics in data collection are described in *Supplementary file 1*.

## Structure determination

The structure determination was performed by the Molecular Replacement method using the α-NPG-bound D59C MelB$_{St}$ mutant structure [PDB ID 7L17] as the search template, followed by rounds of manual building and refinement to resolutions of 2.60 Å, 3.05, 3.45, or 3.68 Å for structure with α-NPG, melibiose, raffinose, or α-MG, respectively, in Phenix (*Liebschner et al., 2019*). The model building and refinement were performed in Phenix and Coot (*Casañal et al., 2020*), respectively. The structures were modeled from positions 2–453 or 455, respectively, without gaps, and the missing side chains due to density disorder, as well as the Ramachandran assessment, are listed in *Supplementary file 5*.

## Sugar docking and modeling

One strong positive density, with varying sizes and shapes, was observed in the difference maps of each of the four structures. The size and shape matched the sugars that co-crystallized, and the docked sugar molecules fitted well with the densities. The sugar refinement restraints were generated from SMILES using the ELBOW program in Phenix (*Liebschner et al., 2019*). To the 2.60 Å α-NPG-bound map, five water molecules were modeled. In addition, a PEG molecule (ligand ID 1PE) was also modeled to a strong positive density with a sausage shape aligning with helix IX. To the melibiose-bound structure, two water molecules were added.

## Hydrogen-deuterium exchange coupled to mass spectrometry (HDX-MS)

An in-solution HDX-MS experiment was performed to study the substrate-induced structural dynamics of MelB$_{St}$. As described (*Hariharan et al., 2024b*), the labeling, quenching, lipid removal, and online digestion were achieved using a fully automated manner using an HDx3 extended parallel system (LEAP Technologies, Morrisville, NC; *Hamuro et al., 2003*; *Hamuro and Coales, 2018*). MelB$_{St}$ was prepared at 50.0 μM in a Na$^+$-free buffer (25 mM Tris-HCl, pH 7.5, 150 mM choline chloride, 10% glycerol, 0.035% UDM in H$_2$O), either in the absence of a substrate (apo) or in the presence of 100 mM melibiose, 100 mM NaCl, or both. The hydrogen/deuterium exchange reaction, as described previously (*Hariharan et al., 2024b*). Briefly, aliquots of 4 μl of each sample were diluted 10-fold into the labeling buffer (25 mM Tris-HCl, pD 7.5, 50 mM choline chloride, 10% glycerol, 0.035% UDM in D$_2$O), without or with 100 mM of melibiose, Na$^+$, or both. Labeled samples were incubated in D$_2$O buffer at 20 °C for multiple time points (30 s, 300 s, and 3000 s) in triplicate, and non-deuterated controls were prepared similarly, except that H$_2$O buffer was used in the labeling step.

At each designated time point, the reaction was quenched by adding an equal volume of ice-cold quench buffer (6 M urea, 100 mM citric acid, pH 2.3 in $H_2O$) for 180 s at 0 °C and immediately subjected to a lipid filtration module integrated on the LEAP PAL system. After incubation of a 60 s with ZrO2 particles, the LEAP X-Press then compressed the filter assembly to separate proteins from the ZrO2 particles-bound phospholipids and detergents. The filtered protein sample was injected into a cool box for online digestion and separation.

LC/MS bottom-up HDX was performed using a Thermo Scientific Ultimate 3000 UHPLC system and Thermo Scientific Orbitrap Eclipse Tribrid mass spectrometer. Samples were digested with a Nepenthesin-2 (Affipro, Czech Republic) column at 8 °C and then trapped in a 1.0 mm x 5.0 mm, 5.0 μm trap cartridge for desalting over 180 sec. The resulting peptides were then separated on a Thermo Scientific Hypersil Gold, 50x1 mm, 1.9 μm, C18 column with a gradient of 10 % to 40 % gradient (A: water, 0.1% formic acid; B: acetonitrile, 0.1% formic acid) for 15 minutes at a flow rate of 40 μL/min. A pepsin wash was added in between runs to minimize the carryover.

A nonspecific digested peptide database has been created for MelB$_{St}$ with a separate MS/MS measurement of non-deuterated samples as described (*Hariharan et al., 2024b*). Digested peptides from undeuterated MelB$_{St}$ protein were identified on the orbitrap mass spectrometer using the same LC gradient as the HDX-MS experiment. Using the Thermo BioPharma Finder software (v 5.1), MS2 spectra were matched to the MelB$_{St}$ sequence with fixed modifications.

A total of 146 or 150 peptide assignments (with confident HDX data across all labeling times) were confirmed for MelB$_{St}$ samples, resulting in 86–87% sequence coverage. The MS data were processed using the Sierra Analytics HDExaminer software with the MelB$_{St}$ peptide database. Following the automated HDX-MS analysis, manual curation was performed. Upon the completion of the data review, a single charge state with high-quality spectra for all replicates across all HDX labeling times was chosen to represent HDX for each peptide. Differential HDX data were tested for statistical significance using the hybrid significance testing criteria method with an in-house MATLAB script, where the HDX differences at different protein states were calculated ($\Delta D = D_{Holo}$ DAp$_o$). Mean HDX differences from the three replicates were assigned as significant according to the hybrid criteria based on the pooled standard deviation and Welch's t-test with $p < 0.05$. The statistically significant differences observed at each residue ($\Delta D_{Mel-Apo} < |0.186|$, $\Delta D_{(Na+)}$ - Apo < $|0.224|$, or $\Delta D_{Na(+)Mel}$ - Apo < $|0.175|$) were used to map HDX consensus effects based on overlapping peptides onto the structure models.

## Statistics and reproducibility

All experiments were performed two to four times. The average values were presented with standard errors. An unpaired t-test was used for statistical analysis. For the relative D%, the data were transferred to log values prior to the unpaired t-test.

## Graphs

Pymol (3.1.5.1) (*Schrodinger, 2013*) and UCSF ChimeraX (*Pettersen et al., 2021*) were used to generate all graphs. The program Origin 2024 was used to plot the ITC curves and transport data.

## MD simulations

The crystal structure of the α-NPG-bound D59C MelB$_{St}$ at a resolution of 2.6 Å was replaced with a melibiose molecule according to the melibiose-bound structure, and the D59C was mutated back to Asp. Both Asp59 and Asp55 residues were set in the deprotonated state. Three systems, including the apo, melibiose-bound, and melibiose- and Na$^+$-bound states, were generated by embedding them in a POPE:POPG (7:2) lipid bilayer created using the CHARMM-GUI (*Wu et al., 2014*) web server. The lipid bilayer was capped with a 25 Å water box on each side, and ~0.15 M NaCl was added to neutralize the system charge and mimic the ionic strength of physiological conditions. The resulting system has ~120,000 atoms, with a periodic boundary condition of ~110 × 110×125 (Å). The ff14SB (*Maier et al., 2015*), lipid17 (*Skjevik et al., 2016*), and GLYCAM (*Group, 2005*) force fields were employed to treat the protein, lipid, and melibiose, respectively. The TIP3P (*Jorgensen et al., 1983*) water model was used for all water molecules.

After system assembly, energy minimization was carried out with harmonic restraints (1000 kJ/mol/Å²) on protein and lipid heavy atoms for 40,000 steps. This was followed by 200 ps of equilibration

in the constant NVT ensemble at 300 K and 1 ns of NPT equilibration with gradually decreased force constants in the harmonic restraints. Five independent replicas of production simulations in the constant NPT ensemble were initiated from different snapshots in the NVT trajectory. For each replica, a trajectory was propagated for ~400 ns at 300 K and 1 atm. The total sampling time for each system was ~2 μs. The temperature was controlled with a Langevin thermostat (*Sindhikara et al., 2009*) with a friction coefficient of 1 $ps^{-1}$ and pressure with a Berendsen barostat (*Uberuaga et al., 2004*) using anisotropic scaling with a relaxation time of 1 ps. Long-range electrostatics were treated with the Particle Mesh Ewald (PME) method (*Darden et al., 1993*) (tolerance $5 \times 10^{-4}$), and van der Waals interactions were truncated using a 12 Å cutoff distance. A 2 fs timestep was used throughout all simulations. The SHAKE algorithm (*Ryckaert et al., 1977*) was employed to constrain the lengths of all hydrogen-containing covalent bonds. All MD simulations were performed with the AMBER24 software package (*Case et al., 2023*).

Water-1 occupancy identified in the α-NPG-bound crystal structure was evaluated by monitoring interactions of water molecules with four coordinating residues. For each frame, a water molecule was defined as located in the sugar-binding pocket if the distances from its oxygen (O) atom to a few neighboring residues and the melibiose molecule satisfied the following rules. First, four pairs of heavy atoms on the melibiose molecule and neighboring residues were defined: O5 on the galactosyl ring and NE2 on Gln372, O5 and OG1 on Thr373, O6 and NE2 on Gln372, as well as O6 and OG1 on Thr373. Second, for each pair of the above-defined heavy atoms, the distance between the water oxygen atom (O) and each of the two heavy atoms in the pair was calculated. Third, out of all four pairs, if at least in one of them both distances were within 4 Å and at least one distance was below 3.5 Å, the water molecule was defined as occupying the sugar-binding site. The water-1 occupancy in the sugar-binding site was then computed as the fraction of frames containing at least one occupying water molecule over the total number of frames in all replicas of the trajectories.

The side-chain heavy-atom root-mean-square fluctuation (RMSF) of all residues in the sugar- and cation-binding pockets was determined using CPPTRAJ.

## Acknowledgements

The authors thank Dr. William Mallard for creating Figure 5a. The X-ray diffraction datasets were collected at ALS BL 5.01 or 5.0.2.

## Additional information

### Competing interests

Yuqi Shi, Rosa Viner: Employee of Thermo Fisher Scientific. The other authors declare that no competing interests exist.

### Funding

| Funder | Grant reference number | Author |
| --- | --- | --- |
| National Institutes of Health | R35 GM153222 | Lan Guan |
| National Institutes of Health | R35 GM150780 | Ruibin Liang |

The funders had no role in study design, data collection and interpretation, or the decision to submit the work for publication.

### Author contributions

Parameswaran Hariharan, Data curation, Formal analysis, Investigation, Writing – review and editing; Yuqi Shi, Data curation, Formal analysis, Investigation, Writing – original draft, Writing – review and editing; Amirhossein Bakhtiiari, Data curation, Formal analysis, Investigation; Ruibin Liang, Lan Guan, Conceptualization, Data curation, Formal analysis, Supervision, Funding acquisition, Writing – original

draft, Writing – review and editing; Rosa Viner, Conceptualization, Supervision, Writing – review and editing

## Author ORCIDs
Parameswaran Hariharan ![ORCID] https://orcid.org/0000-0002-6020-1547
Yuqi Shi ![ORCID] https://orcid.org/0009-0000-0825-7933
Amirhossein Bakhtiiari ![ORCID] https://orcid.org/0000-0002-3979-0973
Ruibin Liang ![ORCID] https://orcid.org/0000-0001-8741-1520
Rosa Viner ![ORCID] https://orcid.org/0000-0003-0550-5545
Lan Guan ![ORCID] https://orcid.org/0000-0002-2274-361X

Reviewer #1 (Public review): https://doi.org/10.7554/eLife.108335.3.sa1
Reviewer #3 (Public review): https://doi.org/10.7554/eLife.108335.3.sa2
Author response https://doi.org/10.7554/eLife.108335.3.sa3

## Additional files

### Supplementary files
Supplementary file 1. Crystallographic data collection, phase, and refinement statistics.

Supplementary file 2. HDX reaction, labeling details, and statistics.

Supplementary file 3. Relative deuterium uptake and uncovered positions of the apo MelBSt.

Supplementary file 4. HDX results at the sugar- and $Na^+$-binding pockets.

Supplementary file 5. Structure information.

Supplementary file 6. MD simulations of Wat-1 occupancy in sugar-bound MelBSt with or without $Na^+$.

MDAR checklist

### Data availability
The x-ray diffraction datasets and models have been deposited to wwPDB under the accession codes 9OLD for the α-nitrophenyl galactoside-bound complex, 9OLI for the melibiose-bound complex, 9OLR for the α-methyl galactoside-bound complex, as well as 9OLP for the raffinose-bound complex of D59C MelBSt.

The following datasets were generated:

| Author(s) | Year | Dataset title | Dataset URL | Database and Identifier |
|---|---|---|---|---|
| Guan L, Hariharan P | 2026 | Crystal structure of alpha-NPG-bound D59C MelBSt | https://doi.org/10.2210/pdb9OLD/pdb | Worldwide Protein Data Bank, 10.2210/pdb9OLD/pdb |
| Guan L, Hariharan P | 2026 | Crystal structure of the melibiose-bound melibiose transporter | https://doi.org/10.2210/pdb9OLI/pdb | Worldwide Protein Data Bank, 10.2210/pdb9OLI/pdb |
| Guan L, Hariharan P | 2026 | Crystal structure of D59C MelB st bound with alpha-methyl galactoside (aMG) | https://doi.org/10.2210/pdb9OLR/pdb | Worldwide Protein Data Bank, 10.2210/pdb9OLR/pdb |
| Guan L, Hariharan P | 2026 | Crystal structure of a raffinose-bound D59C MelB | https://doi.org/10.2210/pdb9OLP/pdb | Worldwide Protein Data Bank, 10.2210/pdb9OLP/pdb |

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
