## [Editor Report · eLife Assessment]

This manuscript presents **useful** insights into the molecular basis underlying the positive cooperativity between the co-transported substrates (galactoside sugar and sodium ion) in the melibiose transporter MelB. Building on years of previous studies, this **convincing** study improves on the resolution of previously published structures and reports the presence of a water molecule in the sugar binding site that would appear to be key for its recognition, introduces further structures bound to different substrates, and utilizes binding and transport assays, as well as HDX-MS and molecular dynamics simulations to further understand the positive cooperativity between sugar and the co-transported sodium cation. The work will be of interest to biologists and biochemists working on cation-coupled symporters, which mediate the transport of a wide range of solutes across cell membranes.

---

## [Referee Report · Reviewer #1 (Public review)]

While the structure of the melibiose permease in both outward and inward-facing forms has been solved previously, there remains unanswered questions regarding its mechanism. Hariharan et al set out to address this with further crystallographic studies complemented with ITC and hydrogen deuterium exchange (HDX) mass spectrometry. They first report 4 different crystal structures of galactose derivatives to explore molecular recognition showing that the galactose moiety itself is the main source of specificity. Interestingly, they observe a water-mediated hydrogen bonding interaction with the protein and suggest that this water molecule may be important in binding.

The results from the crystallography appear sensible, though the resolution of the data is low with only the structure with NPG better than 3Å. Support for the conclusion of the water molecule in the binding site, as interpreted from the density, is given by MD studies.

The HDX also appears to be well done and is explained reasonably well in the revision.

---

## [Referee Report · Reviewer #3 (Public review)]

Summary:

The melibiose permease from *Salmonella enterica* serovar Typhimurium (MelBSt) is a member of the Major Facilitator Superfamily (MFS). It catalyzes the symport of a galactopyranoside with Na⁺, H⁺, or Li⁺, and serves as a prototype model system for investigating cation-coupled transport mechanisms. In cation-coupled symporters, a coupling cation typically moves down its electrochemical gradient to drive the uphill transport of a primary substrate; however, the precise role and molecular contribution of the cation in substrate binding and translocation remain unclear. In a prior study, the authors showed that the binding affinity for melibiose is increased in the presence of Na+ by about 8-fold, but the molecular basis for the cooperative mechanism remains unclear. The objective of this study was to better understand the allosteric coupling between the Na+ and melibiose binding sites. To verify the sugar-recognition specific determinants, the authors solved the outward-facing crystal structures of a uniport mutant D59C with four sugar ligands containing different numbers of monosaccharide units (α-NPG, melibiose, raffinose, or α-MG). The structure with α-NPG bound has improved resolution (2.7 Å) compared to a previously published structure and to those with other sugars. These structures show that the specificity is clearly directed toward the galactosyl moiety. However, the increased affinity for α-NPG involves its hydrophobic phenyl group, positioned at 4 Å-distance from the phenyl group of Tyr26 forms a strong stacking interaction. Moreover, a water molecule bound to OH-4 in the structure with α-NPG was proposed to contribute to the sugar recognition and appears on the pathway between the two specificity-determining pockets. Next, the authors analyzed by hydrogen-to-deuterium exchange coupled to mass spectrometry (HDX-MS) the changes in structural dynamics of the transporter induced by melibiose, Na+, or both. The data support the conclusion that the binding of the coupling cation at a remote location stabilizes the sugar-binding residues to switch to a higher-affinity state. Therefore, the coupling cation in this symporter was proposed to be an allosteric activator.

Strengths:

(1) The manuscript is generally well written.

(2) This study builds on the authors' accumulated knowledge of the melibiose permease and integrates structural and HDX-MS analyses to better understand the communication between the sodium ion and sugar binding sites. A high sequence coverage was obtained for the HDX-MS data (86-87%), which is high for a membrane protein.

The revised manuscript shows clear improvement, and the authors have addressed my concerns in a satisfactory manner. Of note, I noticed two mistakes that should be corrected:

- page 11. Unless I am mistaken, the sentence "In contrast, Na+ alone or with melibiose primarily caused deprotections" should be corrected with "protections". The authors may wish to verify this sentence and also the previous one in the main text.

- Figure 8 displays two cytoplasmic gates (one of them should be periplasmic)

---

## [Author Response]

The following is the authors’ response to the original reviews

**eLife Assessment**
This manuscript presents useful insights into the molecular basis underlying the positive cooperativity between the co-transported substrates (galactoside sugar and sodium ion) in the melibiose transporter MelB. Building on years of previous studies, this work improves on the resolution of previously published structures and reports the presence of a water molecule in the sugar binding site that would appear to be key for its recognition, introduces further structures bound to different substrates, and utilizes HDX-MS to further understand the positive cooperativity between sugar and the co-transported sodium cation. Although the experimental work is solid, the presentation of the data lacks clarity, and in particular, the HDX-MS data interpretation requires further explanation in both methodology and discussion, as well as a clearer description of the new insight that is obtained in relation to previous studies. The work will be of interest to biologists and biochemists working on cation-coupled symporters, which mediate the transport of a wide range of solutes across cell membranes.

We express our gratitude to the associate editor, review editor, and reviewers for their favorable evaluation of this manuscript, as well as their constructive comments and encouragement. Their feedback has been integrated to fortify the evidence, refine the data analysis, and elevate the presentation of the results, thereby enhancing the overall quality and clarity of the manuscript.

A brief summary of the modifications in this revision:

(a) We performed four new experiments: (1) intact cell [^3^H]raffinose transport assay; (2) intact cell *p*-nitrophenol detection to demonstrate α-NPG transport; (3) ITC binding assay for the D59C mutant; and (4) molecular dynamics to simulate the water-1 in sugar-binding site and the dynamics of side chains in the Na^+^- and melibiose-binding pockets. All data consistently support the conclusion draw in this article.

(b) We have added a new figure to show the apo state dynamics (the new Fig. 5a,b) and annotated the amino acid residue positions and marked positions in sugar- or Na^+^-binding pockets.

(c) As suggested by reviewer-3, we have moved the individual mapping of ligand effects on HDX data to the main figure, combined with the residual plots, and marked the amino-acid residue positions.

(d) We have added more deuterium uptake plots to cover all residues in the sugar- or Na^+^-binding pockets in the current figure 7 (previously figure 6).

(e) We have added a new figure 8 showing the positions at the well-studied cytoplasmic gating salt-bridge network and other loops likely important for conformational changes, along with a membrane topology marked with the HDX data. We have added a new figure 9 from MD simulations.

**Reviewer #1:**
While the structure of the melibiose permease in both outward and inward-facing forms has been solved previously, there remain unanswered questions regarding its mechanism. Hariharan et al set out to address this with further crystallographic studies complemented with ITC and hydrogen-deuterium exchange (HDX) mass spectrometry.(1) They first report 4 different crystal structures of galactose derivatives to explore molecular recognition, showing that the galactose moiety itself is the main source of specificity. Interestingly, they observe a water-mediated hydrogen bonding interaction with the protein and suggest that this water molecule may be important in binding.

We thank you for understanding what we've presented in this manuscript.

(2) The results from the crystallography appear sensible, though the resolution of the data is low, with only the structure with NPG better than 3Å. However, it is a bit difficult to understand what novel information is being brought out here and what is known about the ligands. For instance, are these molecules transported by the protein or do they just bind? They measure the affinity by ITC, but draw very few conclusions about how the affinity correlates with the binding modes. Can the protein transport the trisaccharide raffinose?

The four structures with bound sugars of different sizes were used to identify the binding motif on both the primary substrate (sugar) and the transporter (MelB_St_). Although the resolutions of the structures complexed with melibiose, raffinose, or a-MG are relatively low, the size and shape of the densities at each structure are consistent with the corresponding sugar molecules, which provide valuable data for confirming the pose of the bound sugar proposed previously. In this revision, we further refine the α-NPG-bound structure to 2.60 Å. The identified water-1 in this study further confirms the orientation of C4-OH. Notably, this transporter does not recognize or transport glucosides in which the orientation of the C4-OH at the glucopyranosyl ring is opposite. To verify the water in the sugar-binding site, we initiated a new collaborative study using MD simulations. Results showed that Wat-1 exhibited nearly full occupancy when melibiose was present, regardless of whether Na^+^ was bound at the cation-binding site.

As detailed in the Summary, we added two additional sets of transport assays and confirmed that raffinose and α-NPG are transportable substrates of MelB_St_. For α-NPG transport, we measured the end products of the process—enzyme hydrolysis and membrane diffusion of p-nitrophenol released from intracellular α-NPG.

As a bonus, based on the WT-like downhill α-NPG transport activity by the D59C uniporter mutant that failed in active transport against a sugar concentration gradient, we further emphasized that the sugar translocation pathway is isolated from the cation-binding site. The new data strongly support the allosteric effects of cation binding on sugar-binding affinity. Thank you for this helpful suggestion.

A meaningful analysis of ITC data heavily depends on the quality of the data. My laboratory has extensive experience with ITC and has gained rich, insightful mechanistic knowledge of MelB_St_. Because of the low affinity in raffinose and a-MG, unfortunately, no further information can be convincingly obtained. Therefore, we did not dissect the enthalpic and entropic contributions but focused on the Kd value and binding stoichiometry.

(3) The HDX also appears to be well done; however, in the manuscript as written, it is difficult to understand how this relates to the overall mechanism of the protein and the conformational changes that the protein undergoes.

We are sorry for not presenting our data clearly in the initial submission. In this revised manuscript, we have made numerous improvements, as described in the Summary. These enhancements in the HDX data analysis provided new mechanistic insights into the allosteric effects, leading us to conclude that protein dynamics and conformational transitions are coupled with sugar-binding affinity. Na^+^ binding restricts protein conformational flexibility, thereby increasing sugar-binding affinity. The HDX study revealed that the major dynamic region includes a sugar-binding residue, Arg149, which also plays a gating role. Structurally, this dual-function residue undergoes significant displacement during the sugar-affinity-coupled conformational transition, thereby coupling the sugar binding and structural dynamics.

**Reviewer #2:**
This manuscript from Hariharan, Shi, Viner, and Guan presents x-ray crystallographic structures of membrane protein MelB and HDX-MS analysis of ligand-induced dynamics. This work improves on the resolution of previously published structures, introduces further sugar-bound structures, and utilises HDX to explore in further depth the previously observed positive cooperatively to cotransported cation Na^+^. The work presented here builds on years of previous study and adds substantial new details into how Na^+^ binding facilitates melibiose binding and deepens the fundamental understanding of the molecular basis underlying the symport mechanism of cation-coupled transporters. However, the presentation of the data lacks clarity, and in particular, the HDX-MS data interpretation requires further explanation in both methodology and discussion.

We appreciate this reviewer's time in reading our previous articles related to this manuscript.

Comments on Crystallography and biochemical work:(1) It is not clear what Figure 2 is comparing. The text suggests this figure is a comparison of the lower resolution structure to the structure presented in this work; however, the figure legend does not mention which is which, and both images include a modelled water molecule that was not assigned due to poor resolution previously, as stated by the authors, in the previously generated structure. This figure should be more clearly explained.

This figure is a stereo view of a density map created in cross-eye style. In this revision, we changed this figure to Fig. 3 and showed only the density for sugar and water-1.

(2) It is slightly unclear what the ITC measurements add to this current manuscript. The authors comment that raffinose exhibiting poor binding affinity despite having more sugar units is surprising, but it is not surprising to me. No additional interactions can be mapped to these units on their structure, and while it fits into the substrate binding cavity, the extra bulk of additional sugar units is likely to reduce affinity. In fact, from their listed ITC measurements, this appears to be the trend. Additionally, the D59C mutant utilised here in structural determination is deficient in sodium/cation binding. The reported allostery of sodium-sugar binding will likely influence the sugar binding motif as represented by these structures. This is clearly represented by the authors' own ITC work. The ITC included in this work was carried out on the WT protein in the presence of Na^+^. The authors could benefit from clarifying how this work fits with the structural work or carrying out ITC with the D59C mutant, or additionally, in the absence of sodium.

Thank this reviewer for your helpful suggestions. We have performed the suggested ITC measurements with the D59C mutant. The purpose of the ITC experiments was to demonstrate that MelB_St_ can bind raffinose and α-MG to support the crystal structures.

Comments on HDX-MS work:While the use of HDX-MS to deepen the understanding of ligand allostery is an elegant use of the technique, this reviewer advises the authors to refer to the Masson et al. (2019) recommendations for the HDX-MS article (https://doi.org/10.1038/s41592-019-0459-y) on how to best present this data. For example:

All authors value this reviewer's comments and suggestions, which have been included in this revision.

(1) The Methodology includes a lipid removal step. Based on other included methods, I assumed that the HDX-MS was being carried out in detergent-solubilised protein samples. I therefore do not see the need for a lipid removal step that is usually included for bilayer reconstituted samples. I note that this methodology is the same as previously used for MelB. It should be clarified why this step was included, if it was in fact used, aka, further details on the sample preparation should be included.

Yes, a lipid/detergent removal step was included in this study and previous ones, and this information was clearly described in the Methods.

(2) A summary of HDX conditions and results should be given as recommended, including the mean peptide length and average redundancy per state alongside other included information such as reaction temperature, sequence coverage, etc., as prepared for previous publications from the authors, i.e., Hariharan et al., 2024.

We have updated the Table S2 and addressed the reviewer’ request for the details of HDX experiments.

(3) Uptake plots per peptide for the HDX-MS data should be included as supporting information outside of the few examples given in Figure 6.

We have prepared and presented deuterium uptake time-course plots for any peptides with ΔD > threshold in Fig. S5a-c.

(4) A reference should be given to the hybrid significance testing method utilised. Additionally, as stated by Hageman and Weis (2019) (doi:10.1021/acs.analchem.9b01325), the use of P < 0.05 greatly increases the likelihood of false positive ΔD identifications. While the authors include multiple levels of significance, what they refer to as high and lower significant results, this reviewer understands that working with dynamic transporters can lead to increased data variation; a statement of why certain statistical criteria were chosen should be included, and possibly accompanied by volcano plots. The legend of Figure 6 should include what P value is meant by * and ** rather than statistically significant and highly statistically significant.

We appreciate this comment and have cited the suggested article on the hybrid significance method. We fully acknowledge that using a cutoff of P < 0.05 can increase the likelihood of false-positive identifications. By applying multiple levels of statistical testing, we determined that P < 0.05 is an appropriate threshold for this study. The threshold values were presented in the residual plots and explained in the text. For the previous Fig. 6 (renamed Fig. S4b in the current version), we have reported the P value. *, < 0.05; **, < 0.01. (The text for 0.01 was not visible in the previous version. Sorry for the confusion.)

(5) Line 316 states a significant difference in seen in dynamics, how is significance measured here? There is no S.D. given in Table S4. Can the authors further comment on the potential involvement in solvent accessibility and buried helices that might influence the overall dynamics outside of their role in sugar vs sodium binding? An expected low rate of exchange suggests that dynamics are likely influenced by solvent accessibility or peptide hydrophobicity. The increased dynamics at peptides covering the Na binding site on overall more dynamic helices suggests that there is no difference between the dynamics of each site.

The current Table S3 (combined from previous Tables S3 and S4 as suggested) was prepared to provide an overall view of the dynamic regions with SD values provided. For other questions, if we understand correctly, this reviewer asked us to comment on the effects of solvent accessibility or hydrophobic regions on the overall dynamics outside the binding residues of the peptides that cover them. Since HDX rates are influenced by two linked factors: solvent accessibility and hydrogen-bonding interactions that reflect structural dynamics, poor solvent accessibility in buried regions should result in low deuterium uptakes. The peptides in our dataset that include the Na^+^-binding site showed lower HDX, likely due to limited solvent accessibility and lower structural stability. It is unclear what this reviewer meant by "increased dynamics at peptides covering the Na binding site on overall more dynamic helices." We did not observe increased dynamics in peptides covering the Na^+^-binding site; instead, all Na^+^-binding residues and nearby sugar-binding residues have lower degrees of deuteriation.

(6) Previously stated HDX-MS results of MelB (Hariharan et al., 2024) state that the transmembrane helices are less dynamic than polypeptide termini and loops with similar distributions across all transmembrane bundles. The previous data was obtained in the presence of sodium. Does this remove the difference in dynamics in the sugar-binding helices and the cation-binding helices? Including this comparison would support the statement that the sodium-bound MelB is more stable than the Apo state, along with the lack of deprotection observed in the differential analysis.

Thanks for this suggestion. The previous datasets were collected in the presence of Na^+^. In the current study, we also have two Na^+^-containing datasets. Both showed similar results: the multiple overlapping peptides covering the sugar-binding residues on helices I and V have higher HDX rates than those peptides covering the Na^+^-binding residues, even when Na^+^ was present.

(7) Have the authors considered carrying out an HDX-MS comparison between the WT and the D59C mutant? This may provide some further information on the WT structure (particularly a comparison with sugar-bound). This could be tied into a nice discussion of their structural data.

Thank you for this suggestion. Comparing HDX-MS between the WT and the D59C mutant is certainly interesting, especially with the increasing amount of structural, biochemical, and biophysical data now available for this mutant. However, due to limited resources, we might consider it later.

(8) Have the authors considered utilising Li^+^ to infer how cation selectivity impacts the allostery? Do they expect similar stabilisation of a higher-affinity sugar binding state with all cations?

We have shown that Li^+^ also works positively with melibiose. Li^+^ binds to MelB_St_ with a higher affinity than Na^+^ and modifies MelB_St_ differently. It is important to study this thoroughly and separately. To answer the second question, H^+^ is a weak coupling cation with little effect on melibiose binding. Since its pKa is around 6.5, only a small population of MelB_St_ is protonated at pH 7.5. The order of sugar-binding cooperativity is highest with Na^+^, then Li^+^, and finally H^+^.

(9) MD of MelB suggests all transmembrane helices are reorientated during substrate translocation, yet substrate and cotransporter ligand binding only significantly impacts a small number of helices. Can the authors comment on the ensemble of states expected from each HDX experiment? The data presented here instead shows overall stabilisation of the transporter. This data can be compared to that of HDX on MFS sugar cation symporter XylE, where substrate binding induces a transition to the OF state. There is no discussion of how this HDX data compares to previous MFS sugar transporter HDX. The manuscript could benefit from this comparison rather than a comparison to LacY. It is unlikely that there are universal mechanisms that can be inferred even from these model proteins. Highlighting differences between these transport systems provides broader insights into this protein class. Doi: 10.1021/jacs.2c06148 and 10.1038/s41467-018-06704-1.

The sugar translocation free-energy landscape simulations showed that both helix bundles move relative to the membrane plane. This analysis aimed to clarify a hypothesis in the field—that the MFS transporter can use an asymmetric mode to perform the conformational transition between inward- and outward-facing states. In the case of MelB_St_, we clearly demonstrated that both domains move and each helix bundle moves as a unit. So only a small number of helices and loops showed labeling changes. Thanks for the suggestion about comparing with XylE. We have included that in the discussion.

(10) Additionally, the recent publication of SMFS data (by the authors: doi:10.1016/j.str.2022.11.011) states the following: "In the presence of either melibiose or a coupling Na^+^-cation, however, MelB increasingly populates the mechanically less stable state which shows a destabilized middle-loop C3." And "In the presence of both substrate and co-substrate, this mechanically less stable state of MelB is predominant.". It would benefit the authors to comment on these data in contrast to the HDX obtained here. Additionally, is the C3 loop covered, and does it show the destabilization suggested by these studies? HDX can provide a plethora of results that are missing from the current analysis on ligand allostery. The authors instead chose to reference CD and thermal denaturation methods as comparisons.

Thank this reviewer for reading the single-molecule force spectroscopy (SMFS) study on MelB_St_. The C3 loop mentioned in this SMFS article is partially covered in the dataset Mel or Mel plus Na^+^ vs. apo, and there is more coverage in the Na^+^ vs. apo dataset. In either condition, no deprotection was detected. The labeling time point might not be long enough to detect it.

**Reviewer #3:**
Summary:The melibiose permease from *Salmonella enterica* serovar Typhimurium (MelB_St_) is a member of the Major Facilitator Superfamily (MFS). It catalyzes the symport of a galactopyranoside with Na^+^, H^+^, or Li^+^, and serves as a prototype model system for investigating cation-coupled transport mechanisms. In cation-coupled symporters, a coupling cation typically moves down its electrochemical gradient to drive the uphill transport of a primary substrate; however, the precise role and molecular contribution of the cation in substrate binding and translocation remain unclear. In a prior study, the authors showed that the binding affinity for melibiose is increased in the presence of Na^+^ by about 8-fold, but the molecular basis for the cooperative mechanism remains unclear. The objective of this study was to better understand the allosteric coupling between the Na^+^ and melibiose binding sites. To verify the sugar-recognition specific determinants, the authors solved the outward-facing crystal structures of a uniport mutant D59C with four sugar ligands containing different numbers of monosaccharide units (α-NPG, melibiose, raffinose, or α-MG). The structure with α-NPG bound has improved resolution (2.7 Å) compared to a previously published structure and to those with other sugars. These structures show that the specificity is clearly directed toward the galactosyl moiety. However, the increased affinity for α-NPG involves its hydrophobic phenyl group, positioned at 4 Å-distance from the phenyl group of Tyr26, which forms a strong stacking interaction. Moreover, a water molecule bound to OH-4 in the structure with α-NPG was proposed to contribute to the sugar recognition and appears on the pathway between the two specificity-determining pockets. Next, the authors analyzed by hydrogen-to-deuterium exchange coupled to mass spectrometry (HDX-MS) the changes in structural dynamics of the transporter induced by melibiose, Na^+^, or both. The data support the conclusion that the binding of the coupling cation at a remote location stabilizes the sugar-binding residues to switch to a higher-affinity state. Therefore, the coupling cation in this symporter was proposed to be an allosteric activator.Strengths:(1) The manuscript is generally well written.(2) This study builds on the authors' accumulated knowledge of the melibiose permease and integrates structural and HDX-MS analyses to better understand the communication between the sodium ion and sugar binding sites. A high sequence coverage was obtained for the HDX-MS data (86-87%), which is high for a membrane protein.

Thank this reviewer for your positive comments.

Weaknesses:(1) I am not sure that the resolution of the structure (2.7 Å) is sufficiently high to unambiguously establish the presence of a water molecule bound to OH-4 of the α-NPG sugar. In Figure 2, the density for water 1 is not obvious to me, although it is indeed plausible that water mediates the interaction between OH4/OH6 and the residues Q372 and T373.

A water molecule can be modeled at a resolution ranging from 2.4 to 3.2 Å, and the quality of the model depends on the map quality and water location. In this revision, we refined the resolution to 2.6 Å using the same dataset and also performed all-atom MD simulations. All results support the occupancy of water-1 in the sugar-bound MelB_St_.

(2) Site-directed mutagenesis could help strengthen the conclusions of the authors. Would the mutation(s) of Q372 and/or T373 support the water hypothesis by decreasing the affinity for sugars? Mutations of Thr121, Arg 295, combined with functional and/or HDX-MS analyses, may also help support some of the claims of the authors regarding the allosteric communication between the two substrate-binding sites.

The authors thank this reviewer for the thoughtful suggestions. MelB_St_ has been subjected to Cys-scanning mutagenesis (https://doi.org/10.1016/j.jbc.2021.101090). Placing a Cys residue at Gln372 significantly decreased the transport initial rate, accumulation, and melibiose fermentation, with minimal effect on protein expression, as shown in Figure 2 of this JBC article, which could support its role in the binding pocket. The T373C mutant retained most of the WT's activities. Our previous studies showed that Thr121 is only responsible for Na^+^ binding in MelB_St_, and mutations decreased protein stability; now, HDX reveals that this is the rigid position. Additionally, our previous studies indicated that Arg295 is another conformationally important residue. In this version, we have added more HDX analysis to explore the relationship between the two substrate-binding sites with conformational dynamics, especially focusing on the gating salt-bridge network including Arg295, which has provided meaningful new insights.

(3) The main conclusion of the authors is that the binding of the coupling cation stabilizes those dynamic sidechains in the sugar-binding pocket, leading to a high-affinity state. This is visible when comparing panels c and a from Figure S5. However, there is both increased protection (blue, near the sugar) and decreased protection in other areas (red). The latter was less commented, could the increased flexibility in these red regions facilitate the transition between inward- and outward-facing conformations? The HDX changes induced by the different ligands were compared to the apo form (see Figure S5). It might be worth it for data presentation to also analyze the deuterium uptake difference by comparing the conditions sodium ion+melibiose vs melibiose alone. It would make the effect of Na^+^ on the structural dynamics of the melibiose-bound transporter more visible. Similarly, the deuterium uptake difference between sodium ion+melibiose vs sodium ion alone could be analyzed too, in order to plot the effect of melibiose on the Na^+^-bound transporter.

Thanks for this important question. We have added more discussion of the deprotected data and prepared a new Fig. 8b to highlight the melibiose-binding-induced flexibility in several loops, especially the gating area on both sides of the membrane. We also proposed that these changes might facilitate the formation of the transition-competent state. The overall effects induced by substrate binding are relatively small, and the datasets for apo and Na were collected separately, so comparing melibiose&Na^+^ versus Na^+^ might not be as precise. In fact, the Na^+^ effects on the sugar-binding site can be clearly seen in the deuterium uptake plots shown in Figures 7-8, by comparing the first and last panels.

(4) For non-specialists, it would be beneficial to better introduce and explain the choice of using D59C for the structural analyses.

Asp59 is the only site that responds to the binding of all coupling cations: Na^+^, Li^+^, or H^+^. Notably, this thermostable mutant D59C selectively abolishes all cation binding and associated cotransport activities, but it maintains intact sugar binding and exhibits conformational transition as the WT, as demonstrated by electroneutral transport reactions including α-NPG transport showed in this articles, and melibiose exchange and fermentation showed previously. Therefore, the structural data derived from this mutant are significant and offer important mechanistic insights into sugar transport, which supports the conclusion that the Na^+^ functions as allosteric activator.

(5) In Figure 5a, deuterium changes are plotted as a function of peptide ID number. It is hardly informative without making it clearer which regions it corresponds to. Only one peptide is indicated (213-226). I would recommend indicating more of them in areas where deuterium changes are substantial.

We appreciate this comment and have modified the plots by marking the residue position as well as labeled several peptides of significant HDX in the Fig 5b. We also provided a deuteriation map based on peptide coverage (Fig. 5a).

(6) From prior work of the authors, melibiose binding also substantially increases the affinity of the sodium ion. Can the authors interpret this observation based on the HDX data?

This is an intriguing mechanistic question. In this HDX study, we found that the cation-binding pocket and nearby sugar-binding residues are conformationally rigid, while some sugar-binding residues farther from the cation-binding pocket are flexible. We concluded that conformational dynamics regulate sugar-binding affinity, but the increase in Na-binding affinity caused by melibiose is not related to protein dynamics. Our previous interpretation based on structural data remains our preferred explanation; therefore, the bound melibiose physically prevents the release of Na^+^ or Li^+^ from the cation-binding pocket. We also proposed the mechanism of intracellular NA^+^ release in the 2024 JBC paper (https://doi.org/10.1016/j.jbc.2024.107427); after sugar release, the rotamer change of Asp55 will help NA^+^ exit the cation pocket into the empty sugar pocket, and the negative membrane potential inside the cell will further facilitate movement from MelB_St_ to the cytosol.

**Recommendations for the authors:**

**Reviewing Editor Comments:**
(1) It would help the reader if the previous work were introduced more clearly, and if the results of the experiments reported in this manuscript were put into the context of the previous work. Lines 283-296 discuss observations that are similar to previous reported structures as well as novel interpretations. It would help the reader to be clearer about what the new observations are.

Thank you for the important comment. We have revised accordingly by adding related citations and words “as showed previously” when we stated our previous observations.

(2) The affinity by ITC is measured for various ligands, but very few conclusions are drawn about how the affinity correlates with the binding modes. Are the other ligands that are investigated in this study transported by the protein, or do they just bind? Can the protein transport the trisaccharide raffinose? The authors comment that raffinose exhibiting poor binding affinity despite having more sugar units is surprising, but this is not surprising to me. No additional interactions can be mapped to these units on their structure, and while it fits into the substrate binding cavity, the extra bulk of additional sugar units is likely to reduce affinity. In fact, from their listed ITC measurements, this appears to be the trend.Additionally, the D59C mutant utilized here in structural determination is deficient in sodium/cation binding. The reported allostery of sodium-sugar binding will likely influence the sugar binding motif as represented by these structures. This is clearly represented by the authors' own ITC work. The ITC included in this work was carried out on the WT protein in the presence of Na^+^. The authors could benefit from clarifying how this work fits with the structural work or carrying out ITC with the D59C mutant, or additionally, in the absence of sodium. For non-specialists, please better introduce and explain the choice of using D59C for the structural analyses.

Thank you for the meaningful comments. We have comprehensively addressed all the concerns and suggestions as listed in the summary of this revision. Notably, the D59C mutant does not catalyze any electrogenic melibiose transport involved in a cation transduction but catalyze downhill transport location of the galactosides, as shown by the downhill α-NPG transport assay in Fig. 1a. The intact downhill transport results from D59C mutant further supports the allosteric coupling between the cation- and sugar-binding sites.

The binding isotherm and poor affinity of the ITC measurements do not support to further analyze the binding mode since none showed sigmoidal curve, so the enthalpy change cannot be accurately determined. But authors thank this comment.

(3) It is not clear what Figure 2 is comparing. The text suggests this figure is a comparison of the lower resolution structure to the structure presented in this work; however, the figure legend does not mention which is which, and both images include a modelled water molecule that was not assigned due to poor resolution previously, as stated by the authors, in the previously generated structure. This figure should be more clearly explained.

We have addressed these concerns in the response to the Public Reviews at reviewer-2 #1.

(4) I am not sure that the resolution of the structure (2.7 Å) is sufficiently high to unambiguously establish the presence of a water molecule bound to OH-4 of the α-NPG sugar. In Figure 2, the density for water 1 is not obvious to me, although it is indeed plausible that water mediates the interaction between OH4/OH6 and the residues Q372 and T373. Please change line 278 to state "this OH-4 water molecule is likely part of sugar binding".

We have addressed these concerns in the response to the Public Reviews at reviewer-3 #1.

(5) Line 290-296: The Thr121 is not represented in any figures, while the Lys377 is. Their relative positioning between sugar water and sodium is not made clear by any figure.

Thanks for this comment. This information has been clearly presented in the Figs. 7-8. Lys377 is closer to the cation site and related far from the sugar-binding site.

(6) Methodology includes a lipid removal step. Based on other included methods, I assumed that the HDX-MS was being carried out in detergent-solubilized protein samples. I therefore do not see the need for a lipid removal step that is usually included for bilayer reconstituted samples. I note that this methodology is the same as previously used for MelB. It should be clarified why this step was included, if it was in fact used, aka, further details on the sample preparation should be included.(7) A summary of HDX conditions and results should be given as recommended, including the mean peptide length and average redundancy per state alongside other included information such as reaction temperature, sequence coverage, etc., as prepared for previous publications from the authors, i.e., Hariharan et al., 2024.

We have addressed these concerns in the response to the Public Reviews at reviewer-2 #4.

(8) Uptake plots per peptide for the HDX-MS data should be included as supporting information outside of the few examples given in Figure 6.

We have addressed these concerns in the response to the Public Reviews at reviewer-2 #4.

(9) A reference should be given to the hybrid significance testing method utilised. Additionally, as stated by Hageman and Weis (2019) (doi:10.1021/acs.analchem.9b01325), the use of P < 0.05 greatly increases the likelihood of false positive ΔD identifications. While the authors include multiple levels of significance, what they refer to as high and lower significant results, and this reviewer understands that working with dynamic transporters can lead to increased data variation, a statement of why certain statistical criteria were chosen should be included, and possibly accompanied by volcano plots. The legend of Figure 6 should include what P value is meant by * and ** rather than statistically significant and highly statistically significant.

We have addressed these concerns in the response to the Public Reviews at reviewer-2 #4.

(10) The table (S3) and figure (S4) showing uncovered residues is an unclear interpretation of the data; this would be better given as a peptide sequence coverage heat map. This would also be more informative for the redundancy in covered regions, too. In this way, S3 and S4 can be combined.

We have addressed these concerns in the response to the Public Reviews at reviewer-2 #4.

(11) Residual plots in Figure 5 could be improved by a topological map to indicate how peptide number resembles the protein amino acid sequence.

Thanks for the request, due to the figure 6 is big so that we add a transmembrane topology plot colored with the HDX results in Fig. 8c.

(12) The presentation of data in S5 could be clarified. Does the number of results given in the brackets indicate overlapping peptides? What are the lengths of each of these peptides? Classical HDX data presentation utilizes blue for protection and red for deprotection. The use of yellow ribbons to show protection in non-sugar binding residues takes some interpretation and could be clarified by also depicting in a different blue. I also don't see the need to include ribbon and cartoon representation when also using colors to depict protection and deprotection. The authors should change or clarify this choice.

We have moved this figure into the current Fig. 6b as suggested by Reviewer-3. To address your questions listed in the figure legend, the number of results shown in brackets indeed indicates overlapping peptides. What are the lengths of each of these peptides? The sequences of each peptide are shown in Figures 7-8 and are also included in Supplemental Figure S5. Regarding the use of color, both blue and green were used to distinguish peptides protecting the substrate-binding site from other regions. The ribbon and cartoon representations are provided for clarity, as the cartoon style hides many helices.

(13) In Table S5, the difference between valid points and protection is unclear. And what is indicated by numbers in brackets or slashes? Additionally, it should be highlighted again here that single-residue information is inferred from peptide-level data. By value, are the authors referring to peptide-level differential data?

Please review our responses in the Public Reviews at reviewer-2 #5.

(14) Line 316 states a significant difference in seen in dynamics, how is significance measured here? There is no S.D. given in Table S4. Can the authors further comment on the potential involvement in solvent accessibility and buried helices that might influence the overall dynamics outside of their role in sugar vs sodium binding? An expected low rate of exchange suggests that dynamics are likely influenced by solvent accessibility or peptide hydrophobicity? The increased dynamics at peptides covering the Na binding site on overall more dynamic helices suggests that there isn't a difference between the dynamics of each site.

Please review our responses in the Public Reviews at reviewer-2 #5.

(15) Previously stated HDX-MS results of MelB (Hariharan et al., 2024) state that the transmembrane helices are less dynamic than polypeptide termini and loops with similar distributions across all transmembrane bundles. The previous data was obtained in the presence of sodium. Does this remove the difference in dynamics in the sugar-binding helices and the cation-binding helices? Including this comparison would support the statement that the sodium-bound MelB is more stable than the Apo state, along with the lack of deprotection observed in the differential analysis.

Please review our responses in the Public Reviews.

(16) MD of MelB suggests all transmembrane helices are reorientated during substrate translocation, yet substrate and cotransporter ligand binding only significantly impacts a small number of helices. Can the authors comment on the ensemble of states expected from each HDX experiment? The data presented here instead shows overall stabilisation of the transporter. This data can be compared to that of HDX on MFS sugar cation symporter XylE, where substrate binding induces a transition to the OF state. There is no discussion of how this HDX data compares to previous MFS sugar transporter HDX. The manuscript could benefit from this comparison rather than a comparison to LacY. It is unlikely that there are universal mechanisms that can be inferred even from these model proteins. Highlighting differences instead between these transport systems provides broader insights into this protein class. Doi: 10.1021/jacs.2c06148 and 10.1038/s41467-018-06704-1.

Please review our responses in the Public Reviews.

(17) Additionally, the recent publication of SMFS data (by the authors: doi:10.1016/j.str.2022.11.011) states the following: "In the presence of either melibiose or a coupling Na^+^-cation, however, MelB increasingly populates the mechanically less stable state which shows a destabilized middle-loop C3." And "In the presence of both substrate and co-substrate this mechanically less stable state of MelB is predominant.". It would benefit the authors to comment on these data in contrast to the HDX obtained here. Additionally, is the C3 loop covered, and does it show the destabilization suggested by these studies? HDX can provide a plethora of results that are missing from the current analysis on ligand allostery. The authors instead chose to reference CD and thermal denaturation methods as comparisons.

Please review our responses in the Public Reviews.

(18) The main conclusion of the authors is that the binding of the coupling cation stabilizes those dynamic sidechains in the sugar-binding pocket, leading to a high-affinity state. This is visible when comparing panels c and a from Figure S5. However, there is both increased protection (blue, near the sugar) and decreased protection in other areas (red). The latter was less commented, could the increased flexibility in these red regions facilitate the transition between inward- and outward-facing conformations? The HDX changes induced by the different ligands were compared to the apo form (see Figure S5). It might be worth it for data presentation more visible to also analyze the deuterium uptake difference by comparing the conditions sodium ion+melibiose vs melibiose alone. You would make the effect of Na^+^ on the structural dynamics of the melibiose-bound transporter. Similarly, the deuterium uptake difference between sodium ion+melibiose vs sodium ion alone could be analyzed too, in order to plot the effect of melibiose on the Na^+^-bound transporter.

Please review our responses in the Public Reviews.

(19) In Figure 5a, deuterium changes are plotted as a function of peptide ID number. It is hardly informative without making it clearer which regions it corresponds to. Only one peptide is indicated (213-226); I would recommend indicating more of them, in areas where deuterium changes are substantial.

Please review our responses in the Public Reviews.

(20) Figure 6, please indicate in the legend what the black and blue lines are (I assume black is for the apo?)

We are sorry that we did not make it clear. Yes, the black was used for apo state and blue was used for all bound states

(21) From prior work of the authors, melibiose binding also substantially increases the affinity of the sodium ion. Can the authors interpret this observation based on the HDX data?

Please review our responses in the Public Reviews.

Addressing the following three points would strengthen the manuscript, but also involve a significant amount of additional experimental work. If the authors decide not to carry out the experiments described below, they can still improve the assessment by focusing on points (1-21) described above.(22) Have the authors considered carrying out an HDX-MS comparison between the WT and the D59C mutant? This may provide some further information on the WT structure (particularly a comparison with sugar-bound). This could be tied into a nice discussion of their structural data.

Please review our responses in the Public Reviews.

(23) Have the authors considered utilising Li^+^ to infer how cation selectivity impacts the allostery? Do they expect similar stabilisation of a higher-affinity sugar binding state with all cations?

Please review our responses in the Public Reviews.

(24) Site-directed mutagenesis could help strengthen the conclusions. Would the mutation(s) of Q372 and/or T373 support the water hypothesis by decreasing the affinity for sugars? Mutations of Thr 121 and Arg 295, combined with functional and/or HDX-MS analyses, may also help support some of the authors' claims regarding allosteric communication between the two substrate-binding sites.

Please review our responses in the Public Reviews.